# Query-efficient model evaluation using cached responses

**Hayden Helm** [1]  **Ben Johnson** [2]  **Carey E. Priebe** [3]

## Abstract

Evaluating a new model on an existing benchmark is often necessary to understand its behavior before deployment. For modern evaluation frameworks, generating and evaluating a response for all queries can be prohibitively expensive. In practice, responses from previously-evaluated models are often cached – creating a potential opportunity to use this additional information to decrease the number of queries required to accurately evaluate a new model. In this paper, we introduce an approach for predicting benchmark performance that leverages cached model responses based on the Data Kernel Perspective Space (DKPS), a method for quantifying the relationship between models in the black-box setting. Theoretically, we show that DKPS-based methods are query-efficient under certain conditions. Empirically, we demonstrate that DKPS-based methods achieve the same mean absolute error as baselines with a substantially decreased query budget. We conclude by proposing an offline method for selecting a set of queries that maximizes the goodness-of-fit on reference models, improving prediction accuracy over random query selection.

## 1. Introduction

Benchmarks for machine learning methods are widely used to measure the field's progress. While they are not without flaw (Recht et al., 2019), the ubiquity and longevity of early benchmarks such as MNIST (LeCun et al., 2010), CIFAR-10 (Krizhevsky, 2009), and the Iris dataset (Fisher, 1936) attest to their practical utility. In the era of modern generative models, benchmarks have shifted away from evaluating performance on a single, well-defined task and moved towards evaluating performance across a suite of tasks designed to capture a model's overall capability. For example, (Liang et al., 2022) introduced the HELM benchmark suite and leaderboard as an attempt to evaluate language models as comprehensively and transparently as possible. Similar multi-task benchmark suites and leaderboards now exist for embedding models (Muennighoff et al., 2023), image generation (Huang et al., 2025), coding (Chen et al., 2021), and "intelligence" (Chollet, 2019), among others.

Although multi-task evaluations offer a more complete picture of a model's strengths and weaknesses, they are often computationally expensive. Evaluating a model on a modern multi-task benchmark typically requires generating and scoring thousands of responses, making full evaluation increasingly impractical as both model sizes and benchmark scopes grow. At the same time, it is easier than ever to produce new model behavior – via parameter efficient fine-tuning (Han et al., 2024), instruction tuning (Zhang et al., 2025), model merging (Matena & Raffel, 2022), distillation (Gou et al., 2021), prompt engineering (Sahoo et al., 2025), etc. – making it infeasible to evaluate each new variant.

These trends motivate the need for query-efficient alternatives to full benchmark evaluation. In this paper, we address this problem by leveraging information and responses from previously evaluated models under the assumption that it is possible to predict a new model's score by leveraging the similarity between its responses and cached responses from already-scored models on a (small) subset of queries.

**Contribution.** We make two primary contributions.

1. **Theoretical:** We prove that a benchmark prediction method based on the Data Kernel Perspective Space (DKPS) is query-efficient relative to estimating benchmark performance using a subset of queries.
2. **Empirical:** We validate theoretical query-efficiency of DKPS-based methods. As summarized in Table 1, DKPS-based methods can reduce the number of queries required to achieve a given performance by more than $10\times$.

### 1.1. Background & related work

**Low-dimensional representations of language models.** Our work contributes to a growing literature on low-

---

[1]Helivan [2]Jataware [3]Johns Hopkins University; Query efficient benchmarking platform available at https://github.com/helivan-research/quench. Correspondence to: Hayden Helm <hayden@helivan.io>.

*Proceedings of the 43rd International Conference on Machine Learning*, Seoul, South Korea. PMLR 306, 2026. Copyright 2026 by the author(s).

*Table 1.* Mean absolute error (MAE) of benchmark prediction methods under varying query budgets for four HELM-Lite tasks. Results averaged over 1024 random query subsets of size $m$ using all available reference models. Evaluation is under the Leave-One-Family-Out protocol. Lower is better; bold indicates lowest MAE per (task, $m$) pair. Ensemble methods yield the best or near-best performance across all settings.

| | legalbench | | | | medqa | | | | wmt_14 | | | | math | | | |
|---|---|---|---|---|---|---|---|---|---|---|---|---|---|---|---|---|
| Num. queries | 1 | 4 | 16 | 64 | 1 | 4 | 16 | 64 | 1 | 4 | 16 | 64 | 1 | 4 | 16 | 64 |
| Population Mean | 0.093 | 0.093 | 0.093 | 0.093 | 0.131 | 0.131 | 0.131 | 0.131 | 0.049 | 0.049 | 0.049 | 0.049 | 0.234 | 0.234 | 0.234 | 0.234 |
| Sample Score | 0.462 | 0.196 | 0.096 | 0.048 | 0.429 | 0.177 | 0.087 | 0.043 | 0.144 | 0.075 | 0.035 | 0.020 | 0.353 | 0.167 | 0.079 | 0.039 |
| IRT[†] | 0.399 | 0.171 | 0.080 | 0.039 | 0.384 | 0.155 | 0.073 | 0.036 | 0.337 | 0.175 | 0.133 | 0.128 | 0.311 | 0.139 | **0.065** | **0.031** |
| DKPS | 0.090 | **0.068** | 0.051 | 0.034 | **0.117** | 0.097 | 0.066 | 0.039 | **0.039** | **0.027** | 0.019 | 0.016 | 0.147 | 0.105 | 0.082 | 0.070 |
| Ens(DKPS) | 0.090 | **0.068** | 0.051 | 0.034 | **0.117** | 0.097 | 0.065 | 0.038 | **0.039** | **0.027** | **0.019** | 0.015 | 0.147 | 0.105 | 0.080 | 0.061 |
| Ens(DKPS+IRT) | **0.089** | 0.070 | **0.050** | **0.031** | **0.117** | **0.096** | **0.061** | **0.032** | **0.039** | **0.027** | **0.018** | **0.013** | 0.145 | **0.098** | 0.068 | 0.042 |

[†]IRT uses the 1-parameter logistic (Rasch) model (Polo et al., 2024); see Appendix E for details.

dimensional representations of language models. Some approaches construct representations by comparing internal activations across models (Duderstadt et al., 2023; Huh et al., 2024; Horwitz et al., 2025) or by comparing model weights directly (Chen et al., 2025). However, evaluation frameworks typically do not require (or even allow) access to model internals, and instead rely only on model responses. The Data Kernel Perspective Space (DKPS), first introduced in the context of monitoring multi-agent systems (Helm et al., 2024b), addresses this setting by mapping a collection of generative models into a low-dimensional Euclidean space via multidimensional scaling of matrices containing average embedded responses. Prior theoretical work on the DKPS established consistency and concentration of the representations (Acharyya et al., 2024; 2025) and of supervised inference thereon (Helm et al., 2025). These works do not address relative performance for fixed query budgets. Our main contributions are a query-efficiency guarantee showing DKPS-based regression outperforms subset scoring given sufficiently many reference models, and an empirical validation of this property for benchmark score prediction.

**Efficient benchmarking.** Several lines of work aim to reduce the computational cost of benchmark evaluation. Polo et al. (2024) use Item Response Theory to construct benchmark subsets of 100 examples (approx. 1% of original size) that predict performance within 2% error. Vivek et al. (2024) propose clustering examples based on cross-model predictions and selecting cluster centers as "anchor points," achieving accurate rankings with significantly fewer examples. Bean et al. (2025) propose item-centric selection based on cognitive embeddings, improving cold-start performance and cross-family transferability. Li et al. (2024) use reinforcement learning to model dependencies across examples, reducing estimation error by 25-50% via active selection. Perlitz et al. (2024) analyze benchmark design choices and propose ranking algorithms achieving 100× cost reductions with minimal "reliability" loss. These techniques typically require explicit or implicit task-specific structure, model

metadata, access to model internals, or an response-level scoring functions. Relatedly, Ilyas et al. (2022) study *data-models*, which predict a model's output on a particular input as a function of training data composition. Our work targets a related problem: predicting a model's aggregate score across a collection of inputs using embedded responses from previously evaluated models.

Our approach is complementary to prior work in that we leverage cached responses from previously evaluated models, operate purely via black-box access and model response similarity, and assume no structure across tasks or subtasks. Importantly, this enables combining our method with existing techniques – e.g., using DKPS-based methods to predict performance on IRT-selected subsets (Polo et al., 2024) or on "anchor points" (Vivek et al., 2024) – to potentially compound efficiency gains beyond what either approach achieves alone.

### 1.2. Problem statement

Given a generative model $f$, we consider the problem of predicting its score on a benchmark $Q^* = \{q_1, \ldots, q_M\}$. Let $y : \mathcal{F} \times 2^{Q^*} \to [0, 1]$ denote the benchmark scoring function. That is, $y(f, Q^*)$ is the score assigned to $f$ after evaluating it on all queries. We refer to $y(f, Q^*)$ as $y$ when the context is clear.

As described above, full evaluation of $f$ on all queries may be infeasible. Instead, we assume access to previously-evaluated (model, score) pairs $(f_1, y_1), \ldots (f_n, y_n)$ and each model's response to each query $\{\{f_i(q_j)\}_{j=1}^M\}_{i=1}^n$. Given this additional information, our goal is to estimate $y$ with $m \ll M$ queries. That is, we seek to identify an estimate $\hat{y}$ that predicts the full benchmark score of $f$ from its responses to a subset $Q \in 2^{Q^*}$ of the benchmark queries and the information available from the previously-evaluated models.

## 2. The Data Kernel Perspective Space

For our purposes, a model $f \in \mathcal{F}$ is a random mapping from a query space $\mathcal{Q}$ to a response space $\mathcal{X}$. Given $q \in \mathcal{Q}$, model responses $f(q)_1, \ldots, f(q)_r$ are sampled $i.i.d.$ from the distribution $F$. We let $g : \mathcal{X} \to \mathbb{R}^p$ be a fixed embedding function that maps a response to a real-valued vector.

Given models $f_1, \ldots, f_n$ and queries $Q = \{q_1, \ldots, q_m\}$, we let $\bar{X}_i \in \mathbb{R}^{m \times p}$ be the matrix whose $j$th row is the average embedded response from $f_i$ to query $q_j$; or, $\bar{X}_{ij\cdot} = \frac{1}{r} \sum_{k=1}^{r} g(f_i(q_j))_k$. Further, we let $D$ be the pairwise distance matrix with entries $D_{ii'} = ||\bar{X}_i - \bar{X}_{i'}||_F$. We refer to the distribution on embedded responses induced by $f_i(q_j)$ as $F_{ij}$.

Following (Acharyya et al., 2024), the $d$-dimensional Data Kernel Perspective Space (DKPS) representations of the models are defined as the vectors $(\widehat{\psi}_1, \ldots, \widehat{\psi}_n)$ that are a solution to

$$(\widehat{\psi}_1, \ldots, \widehat{\psi}_n) = \text{argmin}_{z_i \in \mathbb{R}^d} \sum_{i,i'}^{n} (||z_i - z_j|| - D_{ii'})^2. \quad (1)$$

DKPS enables treating model-level analysis as a problem in a low-dimensional Euclidean space. We note that the solution to Eq. (1) – and the quality of the representations for a given task – may depend on the choice of query set $Q$ and embedding function $g$. When the context is not clear, we emphasize the dependence on $Q$ by referring to $\psi$ as $\psi(Q)$. In the benchmark setting, the choice of $Q$ is reasonably constrained to elements of $2^{Q^*}$. Finally, we let $\widehat{\Psi}_Q : \mathcal{F} \to \mathbb{R}^d$ be a mapping from the model space to the estimated perspective space under $Q$; that is $\widehat{\Psi}_Q(f_i) = \widehat{\psi}_i(Q)$.

Figure 1 shows the $d = 2$ DKPS of models induced by various choices of $n$ and $m$. Each dot is a model colored by score on the counting and probability subtask from HELM-Lite.

### 2.1. Theoretical properties of the DKPS

For a fixed collection of models, query set, and number of responses, the $d$-dimensional DKPS representations of the models – $\widehat{\psi}_1, \ldots, \widehat{\psi}_n$ – are estimates of a set of "true" $d$-dimensional vectors $\psi_1, \ldots, \psi_n$. That is, as $n$, $m$, and $r$ tend to infinity at particular relative rates, $\lim \widehat{\psi}_i \to \psi_i$ (Acharyya et al., 2024). Further, the error of the worst estimate is bounded above by a constant dependent on the number of models, the number of queries, the number of replicates, and the DKPS dimension: $\max_i ||\widehat{\psi}_i - \psi_i||_2 \leq c(n, m, r, d)$ with high probability (Acharyya et al., 2025). Given its importance to provable query-efficiency, we state this last result in its entirety here:

**Theorem 1.** *[Acharyya et al. (2025) Theorem 2] In our setting, suppose $r = \omega(n^3)$ and $\sup_{ij} \ trace \left( Cov \left( F_{ij} \right) \right) =$*

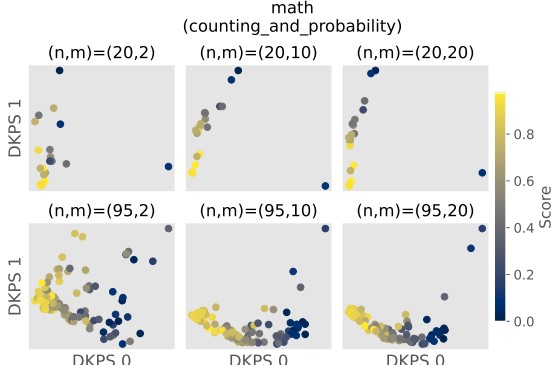

math
(counting_and_probability)

*Figure 1.* Example $d = 2$-dimensional Data Kernel Perspective Spaces (DKPS) for models publicly evaluated on HELM-Lite's MATH counting and probability subtask. Each panel includes the DKPS representations for different $(n, m)$ = (number of models, number of queries) pairs induced by a random query set of size $m$. Each dot is a model colored by its score on the subtask. As the number of queries increases (left to right), the models with similar scores are more tightly clustered. As the number of models increases (top to bottom), there are more models to help localize score signal.

$O(1)$. *Then, under technical assumptions, for every $\delta \in (0, 1/2)$ and for sufficiently large $n$ and $r$*

$$\max_i ||\widehat{\psi}_i - \psi_i||_2 \leq Poly_3 \left( \left( \frac{n^3}{r} \right)^{\frac{1}{2} - \delta} \right)$$

*with high probability, where $Poly_3 (x)$ is a third degree polynomial in $x$.*

The concentration result of Theorem 1 is necessary to make claims related to the quality of representations for a given $(n, m, r)$, which is critical for analyzing the query efficiency of DKPS-based methods for benchmark score prediction.

### 2.2. Inference in the DKPS

Following Helm et al. (2025), we consider model-level inference in DKPS as a statistical problem: Let $(f, y), (f_1, y_1), \ldots, (f_n, y_n)$ be $i.i.d.$ samples from a joint distribution on (model, score) pairs. In the benchmark prediction setting, given $f$ the true benchmark score $y$ is deterministic and analysis with the marginal distribution on $f$ is equivalent to analysis with the joint distribution. We refer to the distribution on models as $P_f$.

Our goal is to construct a decision function $h : \mathcal{F} \to \mathcal{Y}$ that minimizes the expected loss over $P_f$. In practice, operating directly on the space of models is too complex. Instead, we consider the proxy problem of model-level inference in DKPS: we observe perspective-score pairs $(\widehat{\Psi}_Q(f), y), (\widehat{\Psi}_Q(f_1), y_1), \ldots, (\widehat{\Psi}_Q(f_n), y_n)$ and use these to construct a decision function $h_n^Q : \mathcal{F} \to \mathcal{Y}$ defined by $h_n^Q(f) = \hat{h}_n^Q(\widehat{\Psi}_Q(f))$ where $\hat{h}_n^Q : \mathbb{R}^d \to \mathcal{Y}$ is

trained on $\{(\widehat{\Psi}_Q(f_i), y_i)\}_{i=1}^n$. Formally, with loss function $\ell : \mathcal{Y} \times \mathcal{Y} \to \mathbb{R}$, our goal is to select $\hat{h}_n^Q$ to minimize

$$\mathbb{E}_{P_f}[\ell(h_n^Q(f), y)] = \mathbb{E}_{P_f}[\ell(\hat{h}_n^Q(\widehat{\Psi}_Q(f)), y)].$$

We sometimes refer to $h_n^Q$ as $h_n^{(m)}$ as context requires. Note that the responses from the target model $f$ on are included when learning the mapping $\widehat{\Psi}_Q$ but not when learning the decision function.

## 3. Provable query-efficiency in the DKPS

In this section we define query-efficiency and prove that a simple regression function on the perspectives is query-efficient relative to estimating the model's score with its score on a subset of the benchmark queries. Recall $y := y(f, Q^*)$.

For a fixed $Q \subseteq Q^*$ with $|Q| = m$, we say the sequence of decision functions $(h_1^Q, h_2^Q, \ldots)$ is $Q$ query-efficient relative to $(h_1'^Q, h_2'^Q, \ldots)$ if there exists $N \in \mathbb{N}$ such that

$$\mathbb{E}_{P_f}\left[\ell\big(h_n^Q(f),\ y\big)\right] \leq \mathbb{E}_{P_f}\left[\ell\big(h_n'^Q(f),\ y\big)\right] \qquad (2)$$

for all $n > N$. We say the sequence of decision functions $(h_1^{(m)}, \ldots, h_n^{(m)})$ is *m query-efficient* relative to $(h_1'^{(m)}, \ldots, h_n'^{(m)})$ if for all $Q$ with $|Q| = m$ there exists $N_Q \in \mathbb{N}$ such that Eq. (2) holds for all $n > N_Q$.

Finally, we say the sequence of decision functions $(h_1^{(m)}, \ldots, h_n^{(m)})$ is *query-efficient* relative to $(h_1'^{(m)}, \ldots, h_n'^{(m)})$ if for all $m < M$ there exists $N^{(m)} \in \mathbb{N}$ such that it is $m$-query efficient.

For our theoretical analysis we assume the benchmark score function is well-defined on all subsets of $Q^*$. For $Q \subset \mathcal{Q}$, we let $\hat{y}_Q := y(f, Q)$.

Let $h_n^{(m)}$ be nearest neighbor regression in perspective space and $\delta^* = \min_i ||\widehat{\psi}_i - \widehat{\psi}||_2$:

$$\hat{y}_{NN} := h_n^{(m)}(\widehat{\psi}) = \frac{\sum_{i=1}^n \mathbf{1}\{i : ||\widehat{\psi}_i - \widehat{\psi}||_2 = \delta^*\} y_i}{\sum_{i=1}^n \mathbf{1}\{i : ||\widehat{\psi}_i - \widehat{\psi}||_2 = \delta^*\}}.$$

To establish query-efficiency of nearest neighbor regression in DKPS, we require two key assumptions. We let $\text{MSE}(\hat{y}) = \mathbb{E}_{P_f}[(\hat{y} - y)^2]$ denote the mean squared error of the estimator $\hat{y}$ over the model distribution $P_f$.

**Assumption 1** (Lipschitz Score Function). *Given $Q \subseteq 2^{Q^*}$, the benchmark score function $y(\ \cdot\ , Q^*) : \mathcal{F} \to \mathbb{R}$ is $\gamma$-Lipschitz on the perspective space induced by $Q$; or, there exists $\gamma > 0$ such that for any $f, f' \in \mathcal{F}$,*

$$|y(f, Q^*) - y(f', Q^*)| \leq \gamma \cdot ||\psi(Q) - \psi'(Q)||_2.$$

The smoothness on the mapping from $y$ to $\psi(Q)$ ensures that nearby models in DKPS have similar benchmark scores.

In practice, when $m = M$ Assumption 1 will hold if the embedding function $g$ is sufficiently smooth with respect to $y$. With the same condition on $g$, Assumption 1 will hold in practice for $m < M$ in a probabilistic sense.

**Assumption 2** (Model Distribution Support). *The model distribution $P_f$ has non-zero measure on all compact subsets of $\mathcal{F}$. Equivalently, for any target model $f$ and radius $\delta > 0$, there exists $\epsilon > 0$ such that*

$$P_f(B_\delta(f)) \geq \epsilon$$

*where $B_\delta(f) = \{f' \in \mathcal{F} : d_{\mathcal{F}}(f, f') < \delta\}$ for some appropriately defined $d_{\mathcal{F}}$.*

Specifically, $d_{\mathcal{F}}$ should be chosen such that $d_{\mathcal{F}}(f, f') < \delta$ implies that $\Psi_Q(f)$ and $\Psi_Q(f')$ are close in the perspective space. A sufficient condition is that the embedding map $\hat{\Psi}_Q$ is continuous with respect to $d_{\mathcal{F}}$. In practice, Assumption 2 suggests that a large, diverse set of reference models may be needed to realize query efficiency for an arbitrary target model.

Given Assumptions 1 & 2 we are able to arbitrarily bound the prediction error of nearest neighbor regression as a function of $(n, m, r)$. We now state our main theoretical result:

**Theorem 2.** *For any $\epsilon > 0$ there exists $(n, m, r)$ such that $MSE(\hat{y}_{NN}) \leq \epsilon$ with high probability. That is, for $m < M$ such that $MSE(\hat{y}_Q) > 0$, $\hat{y}_{NN}$ is query-efficient relative to $\hat{y}_Q$ with high probability.*

We provide the proof of Theorem 2 in its entirety.

*Proof.* Fix $Q \subseteq Q^*$. Let $f$ be a target model with true perspective $\psi$ and estimated perspective $\widehat{\psi}$, and let $f^* \in \arg\min_i ||\widehat{\psi}_i - \widehat{\psi}||_2$ denote one if its nearest neighbors with perspectives $\psi^*$ and $\widehat{\psi}^*$. The prediction error is $|\hat{y}_{NN} - y| = |y(f^*, Q^*) - y(f, Q^*)|$.

By Assumption 1, we have $|y(f^*, Q^*) - y(f, Q^*)| \leq \gamma \cdot ||\psi^* - \psi||_2$. By the triangle inequality:

$$||\psi^* - \psi||_2 \leq ||\psi^* - \widehat{\psi}^*||_2 + ||\widehat{\psi}^* - \widehat{\psi}||_2 + ||\widehat{\psi} - \psi||_2.$$

Let $\delta^* = ||\widehat{\psi}^* - \widehat{\psi}||_2$ and let $c = c(n, m, r, d)$ be the concentration bound from Theorem 1. With high probability, $\max_i ||\widehat{\psi}_i - \psi_i||_2 \leq c$, so $||\psi^* - \psi||_2 \leq 2c + \delta^*$. Thus $|y^* - y| \leq \gamma(2c + \delta^*)$.

By Theorem 1, for $r = \omega(n^3)$ and sufficiently large $n$ and $r$, the constant $c \leq \text{Poly}_3((n^3/r)^{1/2-\delta})$ can be made arbitrarily small.

By Assumption 2, for any $\epsilon' > 0$, there exists $n_0$ such that for $n > n_0$, $P_f(\min_i ||\psi_i - \psi||_2 < \epsilon') \geq 1 - \eta$ for arbitrarily small $\eta > 0$. Combined with the concentration result, $\delta^* < \epsilon' + 2c$ with high probability.

Therefore, with high probability:

$$|y^* - y| \leq \gamma(2c + \delta^*) < \gamma(4c + \epsilon').$$

For any $\epsilon > 0$, choose $c < \epsilon^{1/2}/(8\gamma)$ and $\epsilon' = \epsilon^{1/2}/(2\gamma)$ with $n$ and $r$ sufficiently large. Then with high probability, $|y^* - y| < \gamma \cdot \epsilon^{1/2}/\gamma = \epsilon^{1/2}$, so $\mathrm{MSE}(\hat{y}_{NN}) = \mathbb{E}_{P_f}[(y^* - y)^2] \leq \epsilon$ with high probability.

Thus, if $\mathrm{MSE}(\hat{y}_Q) > 0$ then there exists $N$ such that for $n > N$, $\mathrm{MSE}(\hat{y}_{NN}) < \mathrm{MSE}(\hat{y}_Q)$, establishing query-efficiency. □

Theorem 2 states that benchmark score prediction using the geometry induced by DKPS using a sufficiently large number of models and cached responses to a subset of benchmark queries is better than using a model's score on the subset of queries. We note that Theorem 1 requires $r = \omega(n^3)$, to ensure the concentration of estimated perspectives around their population counterparts. However, when models are evaluated at temperature 0 – as is the case for a lot of evaluation settings – each model's response to a given query is deterministic, so $\mathbb{E}[g(f(q))] = g(f(q))$ and $r = 1$ suffices for exact recovery of the mean embedded response. The theoretical result is therefore strictly more general than the empirical settings below.

## 4. Empirical query-efficiency in the DKPS

We next provide empirical evidence that DKPS-based prediction methods are query-efficient.

### 4.1. Evaluation set up

**Benchmarks.** We evaluate DKPS-based prediction methods on tasks from the HELM-Lite benchmark suite (Liang et al., 2022). Specifically, we consider four tasks: MATH (Hendrycks et al., 2021), which includes 7 subject-based subtasks; LegalBench (Guha et al., 2023), which includes 5 legal reasoning subtasks; MedQA (Jin et al., 2020), a multiple-choice medical exam dataset; and WMT-14 (Bojar et al., 2014), which includes 5 language-pair translation subtasks. Each task employs a different response-level scoring function, $s : Q^* \to [0, 1]$: correctness for MATH, quasi exact match for LegalBench and MedQA, and BLEU score for WMT-14.

For our purposes, the score for a given subtask is the average response-level score across all queries in that task. The score for a given task is the average subtask score. Full evaluation requires 437 responses for MATH, 2047 responses for LegalBench, 1000 responses for MedQA, 678 responses for WMT-14.'

**Models.** We evaluate models from the HELM-Lite leaderboard that have been scored on each respective task. Be-

cause models are evaluated asynchronously and the benchmark evolves over time, different tasks have different sets of evaluated models. Critically, our method requires that all models be evaluated on the same set of queries (to construct the distance matrix $D$), so we restrict our analysis to maximal subsets of models sharing a common query set within each subtask and task. The task with the fewest evaluated models is LegalBench (93 models). The median and max number of models evaluated on a given task is 95. In total, 95 unique models appear across all tasks. Table 2 in the Appendix lists all models included in at least one evaluation, and Table 3 in the Appendix indicates which models were not evaluated on which tasks.

To ensure our evaluation reflects genuine predictive performance, we adopt a "Leave-One-Family-Out" (LOFO) evaluation protocol. Models are grouped into families based on their base architecture and training procedure (e.g., all Llama variants form one family; see Table 2 for complete groupings). For each evaluation run, we:

(i) Select a held-out model family as the prediction target
(ii) Sample $n$ reference models from the remaining families
(iii) Sample a query subset $Q$ of size $m$ from the benchmark
(iv) Construct $d$-dimensional DKPS representations for the reference models and the held-out model family using their responses to $Q$
(v) Train a decision function $\hat{h}_n$ on $\{(\widehat{\psi}_i, y_i)\}$ for the $n$ reference models to make prediction $\hat{y} = \hat{h}_n^{(m)}(\widehat{\psi})$
(vi) Predict scores for all models in the held-out family

We repeat this protocol 1024 times for each combination of $(n, m)$ and report the mean absolute error (MAE) averaged across all held-out models. The LOFO protocol ensures that predictions do not rely on model family artifacts. As such, our results provide a conservative estimate of real-world query-efficiency.

**Embedding function & DKPS configuration.** We map model responses to vector representations using task-appropriate embedding functions. Unless stated otherwise, for free-form text responses (MATH, WMT-14) we use the sentence embedding model `gemini-embedding-001`, which produces one 3024-dimensional vector per response. For multiple-choice responses (MedQA, LegalBench), we use one-hot encodings in $\mathbb{R}^p$, where $p$ is the number of answer choices.

While the DKPS framework supports multiple responses per query, HELM-Lite requires only a single response per model-query pair, so $r = 1$ throughout. We assume the pairwise distance matrix $D$ is a Euclidean distance matrix and hence solve Eq. (1) with GraSPy's (Chung et al., 2019) implementation of classical multi-dimensional scal-

ing (Torgerson, 1952). We fix $d = 8$ for consistency across experiments, though adaptive methods such as selecting $d$ based on the elbows in the scree plot (Zhu & Ghodsi, 2006) for each task or subtask may yield further improvements.

**Prediction methods.** We compare four prediction methods: two baselines that do not use DKPS representations, and two DKPS-based approaches.

- **Population Mean**: Predicts the benchmark score as the average score of the $n$ reference models: $\hat{y} = \frac{1}{n}\sum_{i=1}^{n} y_i$. This baseline ignores both the target model's responses and the query subset.

- **Sample Score**: Predicts the benchmark score using the target model's average response-level score on the query subset $Q$: $\hat{y}_{sample} = \frac{1}{m}\sum_{q \in Q} s(f(q))$. When $m = M$, this recovers the true benchmark score. This baseline uses the target model's responses but ignores information from reference models.

- **DKPS**: Trains a linear regressor on the DKPS representations of the $n$ reference models to predict benchmark scores: $\hat{y}_{DKPS} = \beta_0 + \beta_1^\top \hat{\psi}$, where $(\beta_0, \beta_1)$ are learned via ordinary least squares on $\{(\hat{\psi}_i, y_i)\}_{i=1}^{n}$. This method leverages cross-model structure but does not directly use response-level scores. We sometimes evaluate various choices of $n$.

- **Ensemble**: Convex combination of Sample Score and DKPS predictions: $\hat{y}_{ensemble} = \alpha \cdot \hat{y}_{sample} + (1-\alpha) \cdot \hat{y}_{DKPS}$. In our experiments, we set $\alpha = m/M$. This weighting reflects our confidence in the sample-based estimate: when $m$ is small, we rely more on DKPS; when $m \approx M$, we rely more on the direct sample score.

All predictions are clipped to $[0, 1]$. We use ordinary least squares (OLS) for the DKPS regressor due to its strong performance. Empirically, $k$-NN in DKPS exhibits the same qualitative trends but OLS achieves lower MAE across all tasks and query budgets (see Appendix B).

### 4.2. Results

**DKPS-based methods outperform Sample Score at low query budgets.** Figure 2 shows the MAE for each of the prediction methods for the representative subtasks. For $m$ small, DKPS-based methods outperform Sample Score acrosss all subtasks and all number of reference models. Importantly, for $n = \text{ALL}$, the earliest intersection between the performance of $y_{DKPS}$ and $y_{SS}$ is $m \approx 10$ (for MATH's counting and probability). For the other three subtasks, however, the region in which $y_{DKPS}$ outperforms $y_{NN}$ is even more substantial – the performances of the two methods intersect beyond $m = 30$.

The more reference models that are included, the better the DKPS-based methods perform. The effect of adding more reference models depends on the task. As hinted by Assumption 2, it is possible that including more reference models would meaningfully shift the point where the performances intersect. We investigate the effect of a particular reference collection at $n = 20$ in Appendix C.

Table 1 shows performance of the methods as a function of query budget $m$ for the full tasks from HELM-Lite. We observe similar relative performance at the task level as we do at the subtask level.

**Ensemble method dominates across all query budgets.** While DKPS and Sample Score each excel in different number of query regimes, the Ensemble method achieves the best performance across nearly all $(m, \text{task})$ combinations, often outperforming both components and the Population Mean baseline. The ensemble's adaptive weighting ($\alpha = \frac{m}{M}$) is effective: at small $m$, it inherits DKPS's ability to exploit cross-model structure; at large $m$, it transitions to rely on the target model's actual response-level scores. This result has important implications. In particular, practitioners need not choose between methods. In some cases where cross-task structure can be assumed, more adaptive weightings may provide additional efficiency (Helm et al., 2024a) – though the proposed weighting appears sufficient for out-of-the-box use.

**Embedding function choice can have non-negligible effect.** The preceding results used `gemini-embedding-001` to map responses to vectors. Figure 3 examines whether this choice matters by comparing six embedding functions on the Math (counting and probability) subtask. At small $m$, embedding choice has a meaningful effect: the best-performing embedding (`gemini-embedding-001`) achieves roughly 20% lower MAE than the worst (`all-minilm-l6-v2`) at $m = 1$. This gap narrows as query budget increases, and by $m \geq 30$ all embeddings perform comparably.

Notably, embedding capacity does not predict performance. Larger models do not uniformly outperform smaller ones, suggesting that alignment between the embedding space and task structure may matter more than raw dimensionality. For practitioners, we recommend selecting an embedding function carefully when operating at low query budgets: a 2-3% reduction in MAE can translate to significant cumulative savings when evaluating new model variants. We investigate the effect of other pipeline hyperparameters such as the MDS embedding dimension $d$ and the ensemble weight $\alpha$ in Appendix D.

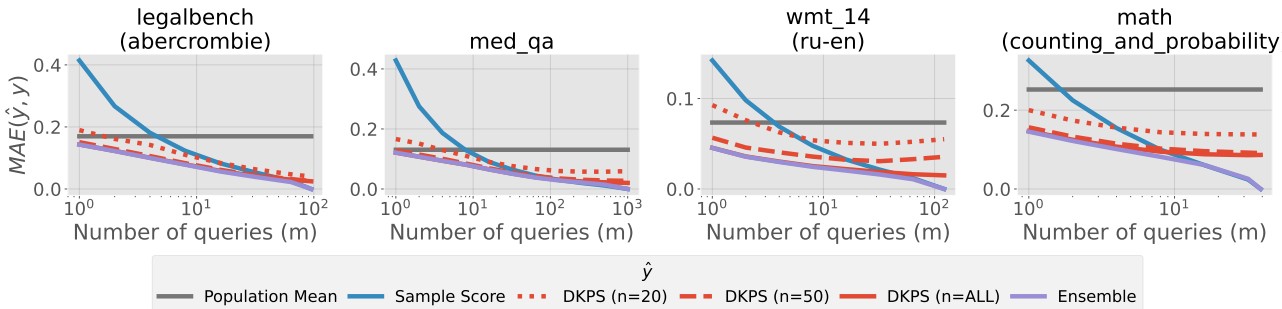

*Figure 2.* Regression in the Data Kernel Perspective Space (DKPS) provides query-efficient benchmark prediction relative to using the sample score across the representative HELM-Lite subtasks. Lines represent the average mean absolute error across leave-one-family-out and 512 randomly sampled query sets. Lower is better. Actual query-efficiency depends on the number of models used to induce DKPS and train the regression function, as well as the task. The ensemble regressor dominates for nearly all number of queries and all tasks.

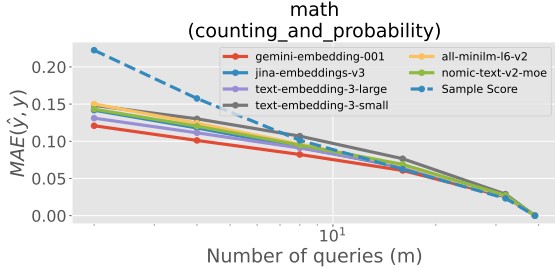

*Figure 3.* Choice of embedding function can have a large effect at small $m$. For small $m$, the best performing embedding model (`gemini-embedding-001`) improves upon the worst performing (`all-minilm-l6-v2`) by $\approx 20\%$ (from MAE $\approx 0.15$ to MAE $\approx 0.12$) at $m = 1$. For large enough $m$, any modern sentence embedding function is sufficient.

**Model-level performance reveals broad applicability**
The top row of Figure 4 shows the distribution of performance gain from using Ensemble (relative to Sample Score) at the individual model level. A dot above $0$ indicates that using the ensemble improve benchmark estimation for that model. As can be seen in the figure, the majority of the mass of each distribution is above $0$ for all $m$ and all subtasks. At low query budgets, the distributions are heavily concentrated above zero across all tasks. This observation indicates that combining DKPS-based prediction with sample scores benefits diverse model types, even with LOFO protocol.

However, the variance in these benefits differs substantially across tasks. WMT exhibits remarkably low spread, with models clustering tightly around the mean, suggesting the Ensemble's adaptive weighting provides uniform improvements for this task. In contrast, MedQA and Math performance differences have wider distributions – some models gain substantially from the Ensemble while others show minimal effects. We leave deeper investigations into model-specific benefits, such as predicting the suitability of DKPS-based methods for a particular model, to future work.

**Query set matters.** The bottom row of Figure 4 shows performance variation across different random query subsets. Each point represents the mean performance difference (Sample Score MAE minus Ensemble MAE) for a single randomly-sampled query set of size $m$, averaged across models. A point above $0$ indicates that the Ensemble outperforms Sample Score for that query set. As can be seen in the figure, at small query budgets ($m \leq 10$), the distributions span a wide range: some query sets yield substantial Ensemble improvements while others produce negligible or negative gains. This variance arises because the quality of the DKPS representations $\widehat{\psi}$ depends on which queries are used to construct the distance matrix $D$ – with few queries, unlucky selections can produce uninformative representations. As query budget increases, the variance contracts, but poor query selection at low $m$ is precisely where DKPS-based methods would otherwise provide the largest savings. This motivates developing query selection strategies that move beyond uniform random sampling.

**Active query set selection protects against bad query sets.**
We propose a simple offline selection strategy that leverages cached responses from reference models. Given a query budget $m$:

  (i) Sample $B$ candidate query sets of size $m$ uniformly at random
 (ii) For each candidate, construct DKPS representations of the reference models and fit a linear regressor to predict their known benchmark scores
(iii) Compute the goodness-of-fit ($R^2$) between predicted and actual scores
 (iv) Select the query set that maximizes $R^2$

If a query set produces DKPS representations that predict benchmark scores for reference models, it likely captures task-relevant model structure and will generalize to new models. This process operates entirely on cached responses and can be performed once and reused for all future evalua-

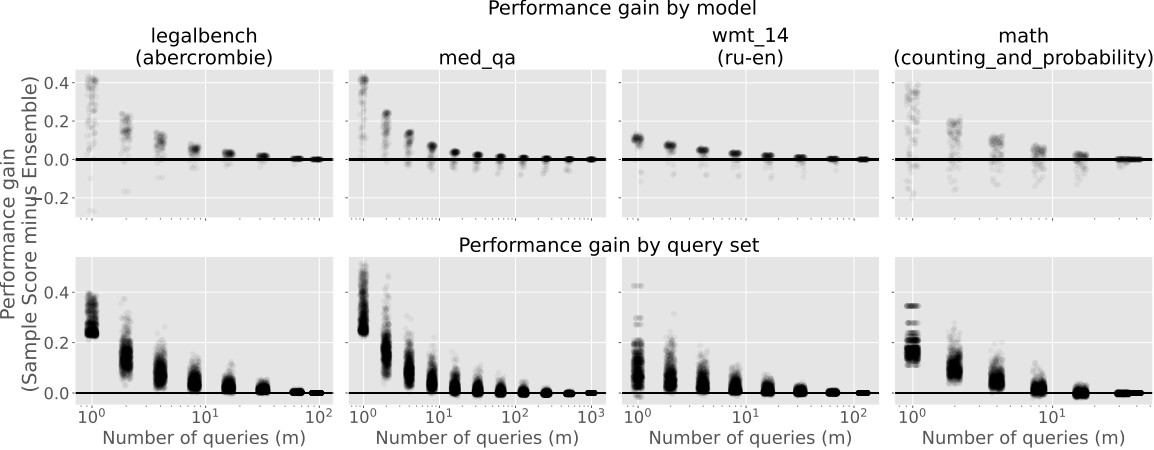

*Figure 4.* Performance gain (MAE of Sample Score minus MAE of Ensemble regressor) on a per model basis (top) and a per query set basis (bottom) for the four representative subtasks. Each dot represents the average difference in performance across query sets (top) or across models (bottom). A A dot above 0 indicates that the Ensemble regressor is better than just using Sample Score. The majority of the mass of the distribution of the difference is above 0 for all number of queries and all tasks.

tions at budget $m$.

Figure 5 demonstrates the effectiveness of this approach on the MATH counting and probability subtask with $m = 8$ queries. The top-left panel shows that $R^2$ on reference models correlates with prediction error on held-out models, validating $R^2$ as a selection criterion. The max-$R^2$ query set (red ×) achieves lower error than the mean random query set. As can be seen in the top-center and bottom panels, this improvement is broad: the selected query set lies in the favorable tail of the error distribution and provides lower error across nearly all individual models. The top-right panel shows that the benefit of active selection diminishes as query budget increases—at larger $m$, random selection suffices. This is expected: active selection matters most where query set variance most limits DKPS effectiveness.

More broadly, and as seen in Table 1 via comparison to IRT (Polo et al., 2024), DKPS-based prediction is complementary to other efficient benchmarking or query selection methods. Our $R^2$-based approach is one such method, but DKPS can equally operate on queries selected via other principled or brute force selection strategies—potentially compounding efficiency gains beyond what either approach achieves alone.

## 5. Discussion

We proposed an approach to benchmark score prediction that leverages cached responses from previously-evaluated models via the Data Kernel Perspective Space. We established formal query-efficiency guarantees, proving that nearest neighbor regression in DKPS outperforms predicting benchmark scores using the model's score on a subset of queries under certain technical assumptions.

These theoretical predictions were validated empirically across diverse HELM-Lite tasks spanning mathematical reasoning, legal analysis, medical question answering, and machine translation. The Ensemble method – which adaptively combines DKPS-based prediction with direct sample scores – consistently achieved the best performance. We further demonstrated that offline query selection strategies can provide additional improvements to query efficiency. Overall, our results demonstrate that information encoded in cached responses enables more efficient performance prediction—an important capability as evaluation continues to scale in cost and complexity.

## Limitations and future work

**Stochastic scoring functions.** Our theoretical analysis assumes the mapping from a response to its score is deterministic. However, many modern evaluation frameworks use LLM-as-judge scoring (Zheng et al., 2023), where scores are stochastic and may depend on contextual factors beyond individual responses. If each score from an LLM-as-a-judge takes the form $s(f(q)) + \epsilon$ with $\epsilon$ i.i.d., then repeatedly querying and scoring recovers the current setting in the limit and extending our framework to this setting is conceptually and mechanistically straightforward.

**Evaluation without response-level scores.** A subtle but important feature of DKPS-based methods is that they remain valid even when response-level scoring is unavailable or expensive. Traditional subset scoring methods require scoring each response to identify informative queries, but DKPS constructs representations purely from response embeddings without invoking the scoring function. This enables prediction in scenarios where: (1) scoring is proprietary or access-controlled (e.g., human evaluation), (2) scoring is

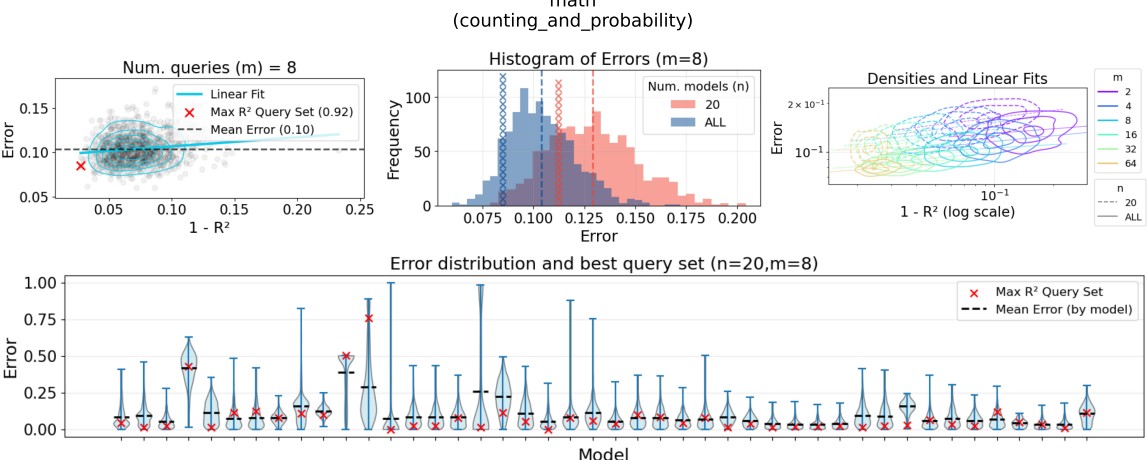

*Figure 5.* Active query selection can improve query-efficiency of DKPS-based prediction methods. **Top left.** Relationship between MAE and linear goodness-of-fit ($R^2$) between DKPS representations of reference models and full benchmark score for $m = 8$ queries on the MATH counting and probability subtask. The highest $R^2$ (lowest $1 - R^2$ is highlighted with a red $\times$. **Top center.** Histogram of MAE for different query subsets. Color indicates number of reference models. Using the query set that induces the DKPS representations that maximizes $R^2$ has a lower better than the average query set for both $n = 20$ and $n = ALL$. **Top right.** MAE versus $1 - R^2$ densities for different $(n, m)$ pairs. **Bottom.** MAE distribution across query sets for various models. If the red $\times$ is lower than the dotted line, the query set that maximized $R^2$ is preferred over the average query set.

computationally expensive (e.g., running code, simulation), or (3) scores are only available at the benchmark level, not per-query.

**Unstructured query sets and missing data.** Our analysis assumes all reference models are evaluated on the same query set, enabling direct construction of the distance matrix $D$. In practice, benchmark data often exhibits partial overlap – different models may be evaluated on different query subsets due to asynchronous evaluation, budget constraints, or benchmark evolution. Several approaches could extend DKPS to this setting. For example, restricting to models sharing a common query set (our current approach), using matrix completion methods (Candes & Recht, 2008) to estimate distances from incomplete data, or aligning separate DKPS representations via models evaluated on multiple query sets (Priebe et al., 2011).

**Prediction bounds for frontier models.** Nearest-neighbor prediction is inherently bounded by the scores of the reference models, which may limit accuracy when the target model substantially outperforms all references. In practice, this constraint is less restrictive than it may appear: most practitioners are evaluating non-frontier models or verifying that modifications to a model do not degrade performance, rather than operating at the boundary of benchmark performance. Extending the theoretical and empirical analysis to support inductive (extrapolative) prediction would require stronger assumptions on the relationships between the task, the model distribution, and the embedding function.

**Query efficiency in practice.** Our empirical results revealed efficiency heterogeneity across task. Understanding which task properties (e.g., response diversity, scoring function complexity, query difficulty distribution, alignment between embeddings and evaluation metrics) predict when DKPS-based methods excel would enable practitioners to assess applicability before deployment.

In terms of practical deployment, our results suggest a simple workflow: maintain cached responses from evaluated models, use the Ensemble method with offline query selection, and allocate approximately 10% of the full query budget for new model evaluation. Notably, our Leave-One-Family-Out evaluation protocol provides conservative estimates of real-world performance since in practice reference sets may include models from the same family as the target, yielding stronger cross-model signal and further improving query efficiency. As model zoos and benchmark suites continue to grow, integrating DKPS-based prediction into evaluation pipelines will yield substantial cumulative savings.

## Impact Statement

The methods proposed and study in this work reduce the computational cost of benchmark evaluation by up to $10\times$ by leveraging cached responses from previously-evaluated models. Lower evaluation costs democratize access to comprehensive model assessment, enabling researchers and practitioners with limited budgets to make informed deployment decisions. We foresee no negative societal consequences of

this work.

## Acknowledgments

We gratefully acknowledge funding from Defense Advanced Research Projects Agency (DARPA) Artificial Intelligence Quantified (AIQ) award number HR00112520026.

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

# A. Experiment details

*Table 2.* List of model families.

| Family | Count | Models |
|---|---|---|
| 01-ai | 3 | yi-34b, yi-6b, yi-large-preview |
| AlephAlpha | 3 | luminous-base, luminous-extended, luminous-supreme |
| ai21 | 5 | j2-grande, j2-jumbo, jamba-1.5-large, jamba-1.5-mini, jamba-instruct |
| allenai | 1 | olmo-7b |
| amazon | 3 | nova-lite-v1, nova-micro-v1, nova-pro-v1 |
| anthropic | 11 | claude-2.0, claude-2.1, claude-3-5-haiku-20241022, claude-3-5-sonnet-20240620, claude-3-5-sonnet-20241022, claude-3-haiku-20240307, claude-3-opus-20240229, claude-3-sonnet-20240229, claude-instant-1.2, claude-instant-v1, claude-v1.3 |
| cohere | 4 | command, command-light, command-r, command-r-plus |
| databricks | 1 | dbrx-instruct |
| deepseek-ai | 2 | deepseek-llm-67b-chat, deepseek-v3 |
| google | 13 | gemini-1.0-pro-001, gemini-1.0-pro-002, gemini-1.5-flash-001, gemini-1.5-flash-002, gemini-1.5-pro-001, gemini-1.5-pro-002, gemini-1.5-pro-preview-0409, gemini-2.0-flash-exp, gemma-2-27b-it, gemma-2-9b-it, gemma-7b, text-bison@001, text-unicorn@001 |
| meta | 12 | llama-2-13b, llama-2-70b, llama-2-7b, llama-3-70b, llama-3-8b, llama-3.1-405b-instruct-turbo, llama-3.1-70b-instruct-turbo, llama-3.1-8b-instruct-turbo, llama-3.2-11b-vision-instruct-turbo, llama-3.2-90b-vision-instruct-turbo, llama-3.3-70b-instruct-turbo, llama-65b |
| microsoft | 3 | phi-2, phi-3-medium-4k-instruct, phi-3-small-8k-instruct |
| mistralai | 9 | mistral-7b-instruct-v0.3, mistral-7b-v0.1, mistral-large-2402, mistral-large-2407, mistral-medium-2312, mistral-small-2402, mixtral-8x22b, mixtral-8x7b-32kseqlen, open-mistral-nemo-2407 |
| nvidia | 1 | nemotron-4-340b-instruct |
| openai | 9 | gpt-3.5-turbo-0613, gpt-4-0613, gpt-4-1106-preview, gpt-4-turbo-2024-04-09, gpt-4o-2024-05-13, gpt-4o-2024-08-06, gpt-4o-mini-2024-07-18, text-davinci-002, text-davinci-003 |
| qwen | 8 | qwen1.5-110b-chat, qwen1.5-14b, qwen1.5-32b, qwen1.5-72b, qwen1.5-7b, qwen2-72b-instruct, qwen2.5-72b-instruct-turbo, qwen2.5-7b-instruct-turbo |
| snowflake | 1 | snowflake-arctic-instruct |
| tiiuae | 2 | falcon-40b, falcon-7b |
| upstage | 1 | solar-pro-241126 |
| writer | 3 | palmyra-x-004, palmyra-x-v2, palmyra-x-v3 |

*Table 3.* Missing models per task.

| Dataset Split | Present | Missing | Missing Models (by family) |
|---|---|---|---|
| legalbench-subset=abercrombie | 93 | 2 | mistralai: mistral-large-2402, mistral-small-2402 |
| legalbench-subset=corporate_lobbying | 93 | 2 | mistralai: mistral-large-2402, mistral-small-2402 |
| legalbench-subset=function_of_decision_section | 93 | 2 | mistralai: mistral-large-2402, mistral-small-2402 |
| legalbench-subset=international_citizenship_questions | 93 | 2 | mistralai: mistral-large-2402, mistral-small-2402 |

*(continued from previous page)*

| Subtask | Present | Missing | Missing Models (by family) |
|---|---|---|---|
| legalbench-subset=proa | 93 | 2 | mistralai: mistral-large-2402, mistral-small-2402 |
| math-subject=algebra | 95 | 0 | - |
| math-subject=counting_and_probability | 95 | 0 | - |
| math-subject=geometry | 95 | 0 | - |
| math-subject=intermediate_algebra | 95 | 0 | - |
| math-subject=number_theory | 95 | 0 | - |
| math-subject=prealgebra | 95 | 0 | - |
| math-subject=precalculus | 95 | 0 | - |
| med_qa | 95 | 0 | - |
| wmt_14-language_pair=cs-en | 95 | 0 | - |
| wmt_14-language_pair=de-en | 95 | 0 | - |
| wmt_14-language_pair=fr-en | 94 | 1 | ai21: jamba-instruct |
| wmt_14-language_pair=hi-en | 95 | 0 | - |
| wmt_14-language_pair=ru-en | 95 | 0 | - |

## B. Nearest neighbor regression in DKPS

Our theoretical results (Theorem 2) establish query-efficiency for nearest neighbor regression in DKPS. In practice, we use ordinary least squares (OLS) due to its stronger empirical performance. Here we compare OLS to 1-NN and $\sqrt{n}$-NN regression across all four tasks, with $n = $ ALL and $d = 8$. Table 4 reports MAE for both the DKPS-only and Ensemble ($\alpha = m/M$) variants.

*Table 4.* MAE of DKPS and Ensemble methods using OLS, 1-NN, and $\sqrt{n}$-NN regression. All settings use $n = $ ALL and $d = 8$. Lower is better.

| Task | $m$ | DKPS | | | Ensemble | | |
|---|---|---|---|---|---|---|---|
| | | OLS | 1-NN | $\sqrt{n}$-NN | OLS | 1-NN | $\sqrt{n}$-NN |
| LegalBench | 1 | 0.102 | 0.130 | 0.104 | 0.102 | 0.130 | 0.104 |
| | 4 | 0.076 | 0.100 | 0.086 | 0.075 | 0.097 | 0.084 |
| | 16 | 0.051 | 0.073 | 0.069 | 0.049 | 0.064 | 0.061 |
| MedQA | 1 | 0.120 | 0.159 | 0.122 | 0.120 | 0.159 | 0.122 |
| | 4 | 0.094 | 0.120 | 0.103 | 0.094 | 0.118 | 0.102 |
| | 16 | 0.066 | 0.084 | 0.075 | 0.064 | 0.078 | 0.070 |
| WMT-14 | 1 | 0.037 | 0.042 | 0.038 | 0.036 | 0.041 | 0.038 |
| | 4 | 0.026 | 0.033 | 0.033 | 0.026 | 0.032 | 0.032 |
| | 16 | 0.021 | 0.028 | 0.031 | 0.019 | 0.024 | 0.026 |
| MATH | 1 | 0.148 | 0.178 | 0.154 | 0.147 | 0.175 | 0.152 |
| | 4 | 0.112 | 0.146 | 0.122 | 0.105 | 0.133 | 0.112 |
| | 16 | 0.090 | 0.121 | 0.099 | 0.063 | 0.077 | 0.067 |

OLS consistently achieves the lowest MAE across all tasks and query budgets. However, both $k$-NN variants exhibit the same qualitative trends as OLS: performance improves with increasing $m$ and increasing $n$, and all three regressors outperform Sample Score at low query budgets. This validates the theoretical predictions of Theorem 2, which guarantees query-efficiency for nearest neighbor regression, while confirming that the additional structure imposed by OLS is beneficial in the moderate-$n$ regime studied here.

## C. Sensitivity to reference model collection

The results in the main text use all available reference models ($n = $ ALL). In practice, the specific set of reference models available may vary. To characterize how performance depends on the reference model collection, we evaluate the Ensemble method ($\alpha = m/M$, $d = 8$) with $n = 20$ reference models randomly sampled from the available pool. For each target

model and subtask, we draw 10 random reference collections and report summary statistics of the resulting MAE, averaged across subtasks within each task.

*Table 5.* Distribution of Ensemble MAE over random reference model collections with $n = 20$. Statistics are averaged across subtasks and target models within each task.

| Task | Min | Q25 | Median | Q75 | Max |
|---|---|---|---|---|---|
| LegalBench | 0.008 | 0.054 | 0.088 | 0.134 | 0.404 |
| MedQA | 0.007 | 0.069 | 0.113 | 0.168 | 0.397 |
| WMT-14 | 0.002 | 0.024 | 0.043 | 0.071 | 0.147 |
| MATH | 0.009 | 0.078 | 0.127 | 0.201 | 0.427 |

With only $n = 20$ reference models, performance depends substantially on which models are included. The interquartile range spans roughly 0.05–0.13 in MAE depending on the task, and worst-case performance can exceed 0.4. WMT-14 exhibits the tightest distribution, suggesting that translation models are more uniformly informative as references. This underscores the importance of maintaining a large and diverse reference model pool: as shown in Figure 2, increasing $n$ from 20 to ALL substantially reduces both the mean error and its variance.

## D. Sensitivity to DKPS dimension $d$ and interpolation weight $\alpha$

We study the sensitivity of the Ensemble method to two key hyperparameters: the DKPS embedding dimension $d$ and the interpolation weight $\alpha$ used to combine DKPS predictions with sample scores. All results use $n = $ ALL.

**Effect of $d$.** Table 6 reports MAE for the Ensemble method with $\alpha = m/M$ across $d \in \{1, 2, 4, 8, 16, 32\}$.

*Table 6.* MAE of the Ensemble method as a function of DKPS dimension $d$, with $\alpha = m/M$ and $n = $ ALL. Lower is better.

| Task | $m$ | $d = 1$ | $d = 2$ | $d = 4$ | $d = 8$ | $d = 16$ | $d = 32$ |
|---|---|---|---|---|---|---|---|
| LegalBench | 1 | 0.112 | 0.098 | 0.098 | 0.102 | 0.110 | 0.137 |
| | 4 | 0.100 | 0.085 | 0.076 | 0.075 | 0.082 | 0.099 |
| | 16 | 0.068 | 0.053 | 0.049 | 0.049 | 0.052 | 0.064 |
| MedQA | 1 | 0.120 | 0.116 | 0.115 | 0.120 | 0.131 | 0.162 |
| | 4 | 0.110 | 0.105 | 0.098 | 0.094 | 0.097 | 0.116 |
| | 16 | 0.073 | 0.068 | 0.066 | 0.064 | 0.063 | 0.071 |
| WMT-14 | 1 | 0.038 | 0.036 | 0.035 | 0.036 | 0.047 | 0.072 |
| | 4 | 0.025 | 0.024 | 0.024 | 0.026 | 0.032 | 0.044 |
| | 16 | 0.017 | 0.016 | 0.017 | 0.019 | 0.025 | 0.030 |
| MATH | 1 | 0.185 | 0.166 | 0.153 | 0.147 | 0.151 | 0.177 |
| | 4 | 0.135 | 0.120 | 0.107 | 0.105 | 0.108 | 0.122 |
| | 16 | 0.081 | 0.076 | 0.064 | 0.063 | 0.064 | 0.069 |

Moderately sized $d$ (e.g., $d \in \{4, 8\}$) appears sufficient for high-quality prediction across all tasks and query budgets. Performance degrades when $d$ is too small (insufficient expressiveness) or too large (overfitting relative to the number of reference models). The optimal $d$ varies slightly by task — for example, LegalBench and WMT-14 favor smaller $d$, while MATH benefits from $d \geq 4$ — but $d = 8$ provides a robust default.

**Effect of $\alpha$.** Table 7 reports MAE for the Ensemble method with $d = 8$ across $\alpha \in \{m/M, 0, 0.1, 0.5, 0.8, 1\}$. Note that $\alpha = 0$ corresponds to DKPS-only and $\alpha = 1$ corresponds to Sample Score.

While $\alpha = m/M$ is not the best choice for every individual $(m, \text{task})$ combination, it is near-best across all settings and provides a robust default that does not require tuning. At small $m$, the Ensemble appropriately down-weights the noisy sample score; at large $m$, it transitions smoothly toward the direct estimate. In the case of MATH at $m = 16$, larger $\alpha$ values (0.5–0.8) slightly outperform $m/M$, reflecting the relatively high quality of direct score estimates on this task at moderate query budgets.

*Table 7.* MAE of the Ensemble method as a function of interpolation weight $\alpha$, with $d = 8$ and $n = $ ALL. Lower is better.

| Task | $m$ | $m/M$ | 0 | 0.1 | 0.5 | 0.8 | 1 |
|---|---|---|---|---|---|---|---|
| LegalBench | 2 | 0.090 | 0.091 | 0.091 | 0.140 | 0.198 | 0.242 |
|  | 4 | 0.076 | 0.076 | 0.075 | 0.103 | 0.140 | 0.168 |
|  | 16 | 0.050 | 0.051 | 0.050 | 0.056 | 0.069 | 0.080 |
| MedQA | 2 | 0.107 | 0.107 | 0.109 | 0.158 | 0.227 | 0.275 |
|  | 4 | 0.094 | 0.094 | 0.094 | 0.119 | 0.158 | 0.187 |
|  | 16 | 0.065 | 0.065 | 0.064 | 0.067 | 0.079 | 0.090 |
| WMT-14 | 2 | 0.031 | 0.031 | 0.031 | 0.055 | 0.079 | 0.097 |
|  | 4 | 0.026 | 0.027 | 0.026 | 0.040 | 0.056 | 0.068 |
|  | 16 | 0.019 | 0.021 | 0.020 | 0.022 | 0.028 | 0.033 |
| MATH | 2 | 0.126 | 0.128 | 0.123 | 0.139 | 0.183 | 0.224 |
|  | 4 | 0.106 | 0.111 | 0.105 | 0.104 | 0.128 | 0.154 |
|  | 16 | 0.068 | 0.090 | 0.082 | 0.060 | 0.057 | 0.065 |

## E. IRT baseline implementation

We compare against Item Response Theory (IRT) following Polo et al. (2024). We use the 1-parameter logistic (Rasch) model, which models the probability of model $i$ answering query $j$ correctly as

$$P(x_{ij} = 1 \mid \theta_i, b_j) = \frac{1}{1 + \exp(-(\theta_i - b_j))}$$

where $\theta_i$ is the ability parameter of model $i$ and $b_j$ is the difficulty parameter of query $j$. The difficulty parameters $\{b_j\}$ are estimated offline from the full set of reference model responses. Given a target model's responses to a subset $Q$ of queries, the ability parameter $\theta$ is estimated via maximum likelihood, and the predicted benchmark score is derived from the estimated ability.

We use the default parameters provided by the publicly available implementation accompanying Polo et al. (2024). All IRT results follow the same Leave-One-Family-Out protocol and use the same random query subsets as the other methods.

We note that IRT performs well on LegalBench, MedQA, and MATH, where it is competitive with or outperforms Sample Score at all query budgets (Table 1). However, IRT performs substantially worse than all other methods on WMT-14. DKPS-based methods, by contrast, perform well across all tasks.

**DKPS+IRT.** We also evaluate a combined method that appends the estimated IRT ability parameter $\hat{\theta}$ as an additional feature to the DKPS perspective vector before fitting OLS. That is, the regressor operates on the $(d + 1)$-dimensional vector $[\hat{\psi}^\top, \hat{\theta}]^\top$. This gives the regressor access to both behavioral similarity (via DKPS) and a direct performance signal (via IRT). Ens(DKPS+IRT) then forms a convex combination of the Sample Score with the DKPS+IRT prediction: $\hat{y} = \alpha \cdot \hat{y}_{\text{sample}} + (1 - \alpha) \cdot \hat{y}_{\text{DKPS+IRT}}$, with $\alpha = m/M$. As shown in Table 1, Ens(DKPS+IRT) achieves the best or near-best performance in 14 of 16 (task, $m$) settings.

