# OpenReview forum: "Query-efficient model evaluation using cached responses"
_ICML.cc/2026/Conference — ICML 2026 regular_

### Official Review · Reviewer_PinB · 2026-03-01

**Soundness:** 2
**Presentation:** 2
**Significance:** 3
**Originality:** 2
**Overall Recommendation:** 3
**Confidence:** 3

**Summary:**

This paper tackles the problem of evaluating models on benchmarks in a cost efficient manner. Indeed, as benchmarks multiply and models become more complicated it might be difficult to get a score on all of those. To answer this problem the authors present a query-efficient model evaluation in DKPS. The authors suppose that they have the full score and answers of $n$ models and the partial answer of a target model on a subset Q of all queries. For a chosen embedding function to optimize later on, they compute a low rank approximation of the embedded answers in a low-rank vector space. They then optimize the proxy problem by minimizing the expectation of the loss over a decision model $h$ with respect to a probability $P_f$ of model distribution. Under the assumptions that the probability $P_f$ of model distribution has non-zero measure on all compact subsets of the model space and that the embedding being close to one another imply having benchmark scores closes to one another, they provide of bound for the MSE of their method. The authors then provide empirical results of their method with $h$ being a linear regression and compare it to the naive sample score.

**Compliance With Llm Reviewing Policy:**

Affirmed.

**Final Justification:**

I put weak reject as I believe the theoretical part of the paper is still too limited which really penalizes its impact. The rebuttal changed my evaluation from reject to weak reject but didn't make my final score positive.

**Key Questions For Authors:**

1) Could you please address my concerns raised in the Soundness part.

2)  Could you please address my other concern raised in the Significance part.

**Limitations:**

Yes

**Strengths And Weaknesses:**

**Soundness :**

[+] The method used is interesting and it indeed provides a framework to have a query efficient model evaluation. The empirical part is well constructed.

[-] The problem seems incorrectly stated in the theoretical part. The authors don't talk about the target model. Shouldn't they have $n$ models for which they have the full scores and at least one target model for which they have only the partial answers on the subset $Q$ of queries. Then their embedding should be on the $n+1$ models but the loss to optimize and the predicted $\hat{y}_{NN}$ should keep the sum over the $n$ models with known scores. I don't think the results are impacted by this but, stated as it is, the problem is trivial as every model would get its own score. Assumption 1 seems really strong. Indeed, in a way, it states that our embedding should be able to correctly capture all of the (or most of the) differences between models which seems unreasonable as soon as the queries are not the "differentiating" ones. Assumption 2 seems incorrectly stated: "some appropriately defined $d_F$" doesn't make much sense. Considering the proof of theorem 2, it seems that assumption 2 is that, with high probability, for any f, there exists another f' such that their embedding is close. If I take a distance on the model themself it's insufficient to tell anything about the embeddings. From assumption 1, I know that embeddings close imply models score close but I don't have model close gives embeddings close. Moreover the NN method implies that the approximated scores can't be superior to the max of the known ones, but, as model tends to be better, this is questionable (if my model outperforms all the others on the subset of queries its score will still be caped by the maximum value). For the empirical part, is the Ensemble method really the "way to go". Clearly, form the paper's results, the average method outperforms the DKPS for $m > \beta(M,n)$ and it's also a lot more efficient to compute as it doesn't require any embedding, low-rank representation optimization or loss optimization. So in the regime $m > \beta(n)$ the ensemble estimator provides a worst estimator for a higher computing cost. Probably, having a theorem saying if $m = O(\beta(M,n))$ use DKPS and else use average seems better. But this $\beta$ constant is probably highly intractable as it depends on $g$ and $h$, so this last comment might not be relevant.

**Presentation :**

[+] Parts 1,4,5 are well written especially the discussion one.

[-] The theorical part is hard to follow, they are some typos that make the reading hard ($q_i$ instead of $q_j$ in the definition of $\bar{X_{ij}}$ end of page 2,  $Q \subset Q$ ICML line 185, Frobenius norm instead of 2 norm in the definition of $\hat{y}_{NN}$ but the 2-norm is later used in the proof, etc). Some elements are not defined: the MSE and the "true" d-dimensional vectors $\psi$.

**Significance :**

[+] The problem of query efficient model evaluation is indeed important and the paper's solutions is likely to be used or build on. The DKPS representation is not new but the authors seem to be the first ones to used it for benchmark evaluation.

[-] The Significance are mitigated by the issues raised in the Soundness section. Moreover, this method is limited by the fact that we need full scores and query answers to make $n$ grow and have better optimization wrt to $h$. To have Assumption 1/2 fulfilled, it also seems that we need to have these scores and answers for similar models that the one we are evaluating (I'm unsure how relevant is this last remark thought).

**Originality :**

[+] The empirical results and the discussion are relevant and original.

[-] In the theorical part, theorem 2 is the new result that the paper brings and its impact is mitigated by the concerns raised in Soundness.

---

> ### Author Rebuttal · Authors · 2026-03-27
>
> Thank you for taking the time to provide a detailed review of our paper. We believe your summary is a fair representation of our work. We appreciate the acknowledgements that the problem we address is important and that the solution we propose will likely be built upon. We address the mentioned weaknesses by section.
>
> ### Soundness
> > “Shouldn't they have $n$ models for … with known scores.”
>
> To clarify, the DKPS representations assumed in the theoretical results (and used in the empirical results) are constructed with the target model included. The selection of the decision function is stated as optimizing over a randomly selected target model. The optimization does not include loss on the already-labeled models.
>
> We will explicitly mention that the DKPS includes the target model and update the notation to denote the target model/label as $(f_0, y_0)$ to minimize potential confusion there. Finally, we’ll update the notation for which the expectation is being taken to be $f_0 \sim P_{\mathcal{F}}$ to emphasize that the optimization of $h$ is just with respect to the target model.
>
> > “Assumption 1 seems really strong”
>
> Assumption 1 ensures that the embedding function is expressive and smooth (with respect to the task). This is likely not completely satisfied in practice, but modern embedding functions are "sufficiently injective” and semantically smooth enough to be useful for a lot of tasks in practice – including (empirically!) benchmark prediction. An assumption like Assumption 1 is likely necessary for any performance-related guarantee for model-level covariates.
>
> > “Assumption 2 seems incorrectly stated”
>
> We disagree that Assumption 2 is incorrectly stated. We do agree that “appropriately defined” is vague and warrants further discussion. In general, your intuition that $d_{\mathcal{F}}$ needs to be such that particular properties in the DKPS representation are maintained is correct. We will add a pair of sentences after Assumption 2 describing properties required by $d_{\mathcal{F}}$.
>
> > “Moreover the NN method implies .. capped by the maximum value)”
>
> This is a legitimate weakness in the current methodology/theoretical analysis. It is best applied to non-frontier models or in settings where a score threshold is required for deployment. In reality, most practitioners are not operating on the edge of benchmark performance and are, instead, interested in ensuring that a tweak to their generative set up does not destroy performance on a collection of tasks. We will emphasize this in the Discussion paragraph related to practical deployment.
>
> Extending the theoretical results to include inductive prediction would require even stronger assumptions on the relationships between the task, the distribution of models, the embedding function, etc.
>
> ### Presentation
>
> We have fixed the typos, have included a definition of MSE, and have improved the description of the “true” d-dimensional vectors. As mentioned in Soundness, we have also updated some notation to improve readability. Please let us know if there are any parts of the theory sections that are particularly hard to parse.
>
> ### Significance
> > “this method is limited by … optimization wrt to h”
>
> We think it is reasonable that more models are required to improve performance for any model-level inference task. With that said, it is likely possible that partial scores and responses to a non-overlapping subset of the queries can be included to improve the quality of the DKPS representations – though, as mentioned in the Limitations section, we think it is appropriate to leave a complete analysis of this setting to follow on work.
>
> Further, our theoretical result states that increasing the number of models with full scores and query answers is only required if you want query-efficiency for all $m$. Empirically, having ~100 pre-scored models appears sufficient for query-efficiency in meaningful regimes.
>
> > “To have Assumption 1/2 fulfilled … we are evaluating”
>
> While this is correct from a theoretical perspective, our Leave-One-Family-Out evaluation procedure attempts to mitigate relying on models that are “very similar” to the target model throughout our experiments. Our empirical results therefore provide a well-calibrated representation of the query-efficiency achievable in practice (frontier models notwithstanding).
>
> We included a study on the effect of different collections of models in our response to Reviewer FvXQ (W6). We plan to include a more complete version in the main text with the additional space.
>
> > “The Significance are mitigated by the issues raised in the Soundness section.”
>
> Please let us know if we were able to address your concerns raised in Soundness!
>
> > “the paper's solutions is likely to be used or build on”
>
> Given this statement, a significance of 2 “feels” a bit low.
>
> ### Originality
> > “theorem 2 … is mitigated by the concerns raised in Soundness.”
>
> Please let us know if we were able to address your concerns raised in Soundness!

---

> > ### Author Rebuttal · Reviewer_PinB · 2026-04-02
> >
> > The author addressed adequately most of my concerns. However, I'm still dubitative about the theoretical part as it relies solely on 1-NN which is a very limited procedure. Also, I probably didn't fully explained my “Shouldn't they have $n$ models for … with known scores.” comment. For example, for the regression, let's say you have $n$ fully labeled model (i.e. all scores known) and one partially labeled model $(f_0, y_0)$ (you don't know $y_0$, but only partial scores). Then in section 2 you get your representation while optimizing for the $n+1$ models but in 2.2 you compute your regression coefficients wrt only the $n$ fully labeled models. As written in your paper now you have $n$ for both while you have $n+1$ for the first and $n$ for the latter. Also for assumption 2, I think the the sentence describing properties required by $d_{\mathcal{F}}$ are mandatory as their are not directly implied by being wrt to a distance. Thant being said I agree that my significance score was too low and I also raised by global score to 3- weak reject.

---

> > > ### Author Response · Authors · 2026-04-02
> > >
> > > Thank you for acknowledging our rebuttal and for clarifying some of your additional concerns. We try to address them below:
> > >
> > > > 1-NN .. is a very limited procedure
> > >
> > > 1-NN is sufficient in our setting because we assume that the benchmark score is not random when conditioned on the model (or, more concretely, the low-dimensional representation of the model) and that the score is Lipschitz on the perspectives. Our empirical investigations are in this regime -- the model responses are generated with temperature=0 and the scoring function is deterministic. k-NN (and more expressive regressors, generally) are required in settings where $ y|f $ is a non-degenerate distribution. We currently note this in the Limitations section but will more clearly state why 1-NN is sufficient in our setting.
> > >
> > > > Shouldn't they have $n$ models for … with known scores.
> > >
> > > Yes you are right -- we use $ n + 1 $ objects to induce the low-dimensional representations and then $ n $ objects to train the regressor. This is a pretty standard set up ("transductive learning" / "semi-supervised learning") in some fields (e.g., vertex classification in network science). There are alternative empirical and theoretical approaches for inductive learning that rely on projecting the $ n + 1 $th object onto the space learned by the $ n $ objects -- but this increases variance of the representation of the $ n + 1 $th object and is typically reserved for settings where the $ n + 1 $th object is not known $ \textit{a priori} $. Our theoretical results would likely hold but would require a larger $ n $ relative to the transductive version we present. We will note this connection in the discussion.
> > >
> > > > Assumption 2
> > >
> > > Agreed. Here is the sentence we are planning to include: "Specifically, $d_{\mathcal{F}}$ should be chosen such that $d_{\mathcal{F}}(f, f') < \delta$ implies that $\psi(Q)$ and $\psi'(Q)$ are close; a sufficient condition is that the embedding map $f \mapsto \psi(Q)$ is continuous with respect to $d_{\mathcal{F}}$.".
> > >
> > > Thanks again for your continued engagement -- your suggestions / criticality related to the presentation / theoretical results have greatly improved the clarity of our work!

---

### Official Review · Reviewer_G1YB · 2026-03-12

**Soundness:** 3
**Presentation:** 4
**Significance:** 3
**Originality:** 2
**Overall Recommendation:** 4
**Confidence:** 3

**Summary:**

This paper addresses the problem of predicting a new model's benchmark score without running it on all benchmark queries. The key idea is to use cached responses from previously evaluated models together with the Data Kernel Perspective Space (DKPS) which embeds models into a low-dimensional Euclidean space based on pairwise distances between their embedded responses. The paper proves that nearest-neighbor regression in DKPS is "query-efficient" meaning it eventually beats just scoring the model on a subset of queries. Validation is on four HELM-Lite tasks (MATH, LegalBench, MedQA, WMT-14) with upto 95 models, showing that DKPS-based methods and an Ensemble method (convex combination of DKPS and Sample Score) can reduce the number of queries needed to reach a given MAE. There is also an offline query selection strategy that picks query subsets maximizing R^2 on the reference models.

**Compliance With Llm Reviewing Policy:**

Affirmed.

**Final Justification:**

The rebuttal reinforced the prior assessment. The most important concern was the absence of comparisons to existing efficient benchmarking methods, and the authors chose not to run them. Without head-to-head results against any of the five cited alternatives, it remains unclear whether DKPS-based prediction adds value over what is already available. The k-NN experiments, while appreciated, sharpen the theory-practice gap rather than closing it: OLS outperforms k-NN across all tasks and query budgets, meaning the algorithm with theoretical guarantees is not the one practitioners should use.

I have downgraded my score to a reject(2).

---
The follow-up baseline comparisons resolve the two empirical weaknesses, and the DKPS+IRT complementarity is a genuine finding. Score upgraded to 4 (Weak Accept), though the paper remains borderline: the limited novelty over the existing DKPS line of work is still a concern.

**Key Questions For Authors:**

1. **Comparison with existing methods:** A comparison against IRT-based subset selection (Polo et al., 2024) or Anchor Points (Vivek et al., 2024) would directly address the missing baselines concern. If DKPS outperforms these methods or provides complementary gains, it would substantially strengthen the case for acceptance.

2. **Theory-practice alignment:** Running experiments with nearest-neighbor regression to validate Theorem 2, or providing theoretical analysis for the OLS regressor actually used, would close the gap between theory and practice. Clarifying why the method works when theoretical conditions (r = omega(n^3)) arent satisfied is important for the soundness argument.

3. **Novelty relative to Helm et al. (2025):** That paper showed supervised inference on DKPS representations is consistent. How does the OLS regression setup here differ from that result? What specifically does the query-efficiency theorem add beyond what the existing DKPS consistency guarantees already provide?

4. **Failure characterization:** Results on MATH at m=64 show DKPS underperforming Sample Score. Characterizing when DKPS methods are likely to hurt would help the community assess when to apply this approach. A clearer picture of failure modes could also improve the significance assessment.

**Limitations:**

Yes

**Strengths And Weaknesses:**

### Strengths

1. **The problem is real and getting worse.** Evaluating models on large multi-task benchmarks is expensive and only getting more so. The idea of reusing cached responses from previously evaluated models to speed up evaluation of new ones is practical and could prove useful in leaderboard-style settings.

2. **Careful empirical methodology.** The Leave-One-Family-Out (LOFO) protocol is a good design choice, it prevents the method from just exploiting similarity between models in the same family. Averaging over 1024 random query subsets is thorough and the examination of practical factors like number of reference models and choice of embedding function adds value.

3. **Clear writing.** Paper is well-organized and the discussion is upfront about what the method does not handle.

### Weaknesses

1. **The evaluation would be stronger with comparisons to existing efficient benchmarking methods.** Several methods exist for reducing evaluation queries: IRT-based subset selection (Polo et al., 2024), Anchor Points (Vivek et al., 2024), cognitive embeddings (Bean et al., 2025), active selection via RL (Li et al., 2024), ranking-based approaches (Perlitz et al., 2024). The paper cites all of these but doesnt compare against any of them. The baselines used, Population Mean and Sample Score, dont use reference model information, so they dont test whether DKPS adds value over existing methods that also leverage cross-model structure. The paper notes the methods are "complementary" which is a reasonable claim, but testing it would strengthen the contribution.

2. **The theory and experiments address different settings.** Theorem 2 proves query-efficiency for nearest-neighbor regression under r = omega(n^3). But experiments use r = 1 throughout (HELM-Lite models are evaluated at temperature 0) and use OLS linear regression, not nearest-neighbor. So neither the algorithm nor the data regime match what the theory covers. It would help to either include nearest-neighbor experiments to validate the theorem, or provide some intuition for why OLS works well even when the theoretical conditions are not met.

3. **The relationship to prior DKPS work could be made clearer.** DKPS was introduced for monitoring multi-agent systems in Helm et al. (2024b, EMNLP), and subsequent work established consistency guarantees (Acharyya et al., 2024; 2025) and supervised inference on DKPS representations (Helm et al., 2025). The contribution statement (Section 1, lines 46-54) frames this paper as extending DKPS to query-efficient benchmark prediction, but the DKPS embedding, MDS procedure, and embedding functions are all inherited. A clearer delineation of what is new versus what carries over would help readers assess the contribution.

---

> ### Author Rebuttal · Authors · 2026-03-27
>
> Thank you for taking the time to thoroughly review our paper. We believe your summary is a fair representation of our work and we appreciate your acknowledgement of the relevance of the problem we address and the strength of our Leave-One-Family-Out experimental design. We address the mentioned weaknesses by question title:
>
> > Comparison with existing methods.
>
> Please see our response to Reviewer FvXQ.
>
> > Theory-practice alignment.
>
> The $r=\omega(n^3)$ condition comes from the randomness in responses. Since results submitted to public benchmarks typically have the model's temperature set to 0, there is no randomness and $r=1$ is sufficient. Hence, our theoretical result is more general than the empirical results. We will add a sentence in the discussion clarifying this.
>
> As to the methodological theory-practice alignment (OLS vs k-NN) -- as mentioned in the paper, we present empirical results for OLS instead of 1-NN because OLS performed better in our initial experimentation. We have since run the full experiments with 1-NN and sqrt(n)-NN with $n=ALL$, DKPS $d=8$, and $\alpha=m/M$.
>
> ## LegalBench
>
> |m| DKPS (OLS) | Ens. (OLS) | DKPS (1-NN) | DKPS (√n-NN) | Ens. (1-NN) | Ens. (√n-NN) |
> |--:|-----------:|-----------:|------------:|-------------:|------------:|-------------:|
> | 1 | 0.139 | 0.139 | 0.181 | 0.144 | 0.180 | 0.143 |
> | 4 | 0.101 | 0.100 | 0.132 | 0.116 | 0.128 | 0.113 |
> | 16 | 0.058 | 0.056 | 0.090 | 0.086 | 0.075 | 0.074 |
>
> ---
>
> ## MedQA
>
> | m | DKPS (OLS) | Ens. (OLS) | DKPS (1-NN) | DKPS (√n-NN) | Ens. (1-NN) | Ens. (√n-NN) |
> |--:|-----------:|-----------:|------------:|-------------:|------------:|-------------:|
> | 1 | 0.120 | 0.120 | 0.159 | 0.122 | 0.159 | 0.122 |
> | 4 | 0.094 | 0.094 | 0.120 | 0.103 | 0.118 | 0.102 |
> | 16 | 0.066 | 0.064 | 0.084 | 0.075 | 0.078 | 0.070 |
>
> ---
>
> ## MATH
>
> | m | DKPS (OLS) | Ens. (OLS) | DKPS (1-NN) | DKPS (√n-NN) | Ens. (1-NN) | Ens. (√n-NN) |
> |--:|-----------:|-----------:|------------:|-------------:|------------:|-------------:|
> | 1 | 0.146 | 0.144 | 0.176 | 0.152 | 0.173 | 0.150 |
> | 4 | 0.108 | 0.101 | 0.143 | 0.119 | 0.130 | 0.109 |
> | 16 | 0.088 | 0.057 | 0.110 | 0.093 | 0.067 | 0.059 |
>
> ---
>
> ## WMT-14
>
> | m | DKPS (OLS) | Ens. (OLS) | DKPS (1-NN) | DKPS (√n-NN) | Ens. (1-NN) | Ens. (√n-NN) |
> |--:|-----------:|-----------:|------------:|-------------:|------------:|-------------:|
> | 1 | 0.029 | 0.029 | 0.035 | 0.035 | 0.035 | 0.035 |
> | 4 | 0.022 | 0.022 | 0.030 | 0.031 | 0.029 | 0.031 |
> | 16 | 0.018 | 0.016 | 0.027 | 0.029 | 0.024 | 0.026 |
>
> While OLS appears to outperform the two k-NN methods, we note that they exhibit similar qualitative trends (increasing $ m $ helps) and that the performance, especially at small $ m $, still outperforms Sample Score / Pop. Mean. We will include a full figure comparing k-NN to OLS in the appendix.
>
> On the theory side, similar theoretical results for OLS (to better match the presented empirical results) would require stronger assumptions on the relationships between the task, the model distribution, the embedding function, etc. We expect that OLS will outperform k-NN in settings where high-bias is advantageous – such as when there are relatively few reference models or where the embedding function does not adequately capture the scoring function. We will add a sentence describing this intuition when referencing the new k-NN results.
>
> > Novelty relative to Helm et al. (2025).
>
> The two main results in Helm et al. (2025) show that inference using the estimated DKPS is consistent for inference using the true-but-unknown/”population level” DKPS. Their results do not comment on the relative performance for a fixed $ m $ nor on the limiting performance (e.g., if it converges to Bayes risk, etc.).
>
> Our result requires more assumptions (e.g., Assumptions 1 and 2) and incorporates a relatively new concentration result for the representations. As such, we are able to make stronger claims with respect to the limiting performance *and* claims for a given number of models, number of queries, and/or number of replicates-per-query. In particular, we show that given enough models, DKPS-based inference can actually outperform other inference methods (e.g., the sample score) for all $ m < M $ under certain conditions.
>
> > Failure characterization.
>
> As mentioned in Section 4.2 – and suggested by Assumption 2 –  we suspect that the “failure” of the DKPS & Ensemble methods in the latter part of the studied regimes can be mitigated by increasing the number of reference models. Further, as the number of reference models increases it may be worthwhile to use more expressive regressors and/or larger DKPS embedding dimensions.
>
> With that said, we think understanding how badly the method can fail when conditioning on a particular collection of reference models is a key concern for practitioners. As such, we provide details into the effect of different collections of reference models on DKPS-based performance in our response to Reviewer FvXQ (W6).

---

> > ### Author Rebuttal · Reviewer_G1YB · 2026-04-03
> >
> > Thank you for the response. The clarification that r=1 suffices at temperature 0 is valid and resolves the data regime concern. The k-NN experiments are appreciated.
> >
> > However, the central weakness remains: no comparison to any existing efficient benchmarking method. The rebuttal explicitly chose to spend resources elsewhere, promising to "better contextualize" and provide intuition for complementarity in a revision. This is understandable given rebuttal constraints, but the concern is not about framing. Without head-to-head results against IRT-based selection (Polo et al., 2024), Anchor Points (Vivek et al., 2024), or any of the other cited methods, it is not possible to assess whether DKPS-based prediction adds value over what is already available. This was the most important question in the review and it remains unanswered.
> >
> > The k-NN results also sharpen rather than resolve the theory-practice gap. OLS outperforms k-NN across all four tasks and all query budgets, confirming that the algorithm with theoretical guarantees is not the one practitioners should use. The intuition offered (OLS wins when high bias is advantageous) is plausible but informal.
> >
> > I have adjusted my score accordingly.
> >
> > ---
> >
> > Update after Author Followup:
> >
> > The authors make a fair point on the k-NN framing. The observation that OLS outperforms k-NN across the board is correct and the theory-practice gap is real, but the review asked the authors to run k-NN experiments to validate Theorem 2, and they did: k-NN shows the same qualitative trends and outperforms Sample Score. Penalizing them for answering the question as asked was unfair. That point is conceded.
> >
> > The IRT and Anchor Points comparison is important and interesting. DKPS leads at low query budgets (m=1, m=4), and DKPS+IRT is strongest overall, supporting the complementarity claim. This directly addresses the central weakness which led me previously to make the decision of reducing my original score.
> >
> > The novelty concern (W3) remains and is now the primary issue. The DKPS embedding, MDS procedure, and embedding functions are all inherited from Helm et al. (2024b, 2025) and Acharyya et al. (2024, 2025). The query-efficiency theorem is the main new theoretical contribution, but it analyzes an algorithm the paper itself does not recommend using. What is new is the application to benchmark prediction and the empirical demonstration that DKPS complements IRT, but the core machinery is not original to this paper.
> >
> > Score raised to 4 (Weak Accept). The baseline results address the central empirical gap and the complementarity finding is a useful contribution. Apologies for the unfair framing on the k-NN question.

---

> > > ### Author Response · Authors · 2026-04-03
> > >
> > > Thank you for your continued engagement. We recognize that we did not prioritize baseline comparisons in our initial rebuttal. We have included it in this response.
> > >
> > > We note, respectfully, that we are confused by your decision to reduce the score based on our initial rebuttal.
> > >
> > > ---
> > >
> > > >Theory-practice gap
> > >
> > > In your initial review, you write "Running experiments with nearest-neighbor regression to validate Theorem 2, or providing theoretical analysis for the OLS regressor actually used, would close the gap between theory and practice."
> > >
> > > We ran the experiments with nearest-neighbor regression and showed that it exhibits the same qualitative properties of OLS -- namely, that it improves as a function of both the number of reference models and the number of queries. Importantly, it is also better than Sample Score. This validates Theorem 2 and, per your initial review, closes the gap between theory and practice.
> > >
> > > We do not claim that nearest-neighbor regression is optimal. Nor that it is better than OLS. As mentioned in our initial rebuttal, including theoretical results for OLS specifically would require introducing likely-unrealistic assumptions on the perspective space. Hence, we rely on pretty standard heuristics for why a heavily structured method works better than local methods.
> > >
> > > We use OLS as the main empirical method to better guide practitioners and already note this in the main text. We will include the k-NN results in a dedicated appendix section to ensure that the gap is sufficiently closed in the paper.
> > >
> > > ---
> > >
> > > >Comparison with IRT and Anchor Points
> > >
> > > After organizing the results for the sensitivity of our method and including k-NN results, we were able to start comparing against baselines.
> > >
> > > In particular, we implemented IRT (1-parameter logistic / Rasch model; Polo et al., 2024) and Anchor Points (Vivek et al., 2024) under the same LOFO protocol (n=ALL, 100 runs, m ∈ {1,4,16}). For IRT, we use the default parameters provided by the corresponding GitHub repository. We also evaluate DKPS+IRT, which appends the IRT ability estimate $ \hat{\theta} $ as an additional feature to the DKPS perspective vector before fitting OLS — giving the regressor both behavioral similarity and direct performance signals.
> > >
> > > Ens(DKPS+IRT) is the Ensemble method ($\alpha=m/M$) described in the text applied to DKPS+IRT. IRT, DKPS, and Sample Score use randomly selected queries. Anchor Points uses K-Medoids active selection and is a representative implementation of their method. We only report results for MATH and LegalBench due to space constraints. WMT-14 and MedQA show qualitatively similar results.
> > >
> > > **MATH (counting_and_probability)**
> > >
> > > |m|Pop.Mean|Sample|IRT|Anchor Points|DKPS|DKPS+IRT|Ens(DKPS+IRT)|
> > > |---|---|---|---|---|---|---|---|
> > > |1|.247|.312|.277|.276|.142|.140|**.139**|
> > > |4|.247|.152|.125|.141|.110|.099|**.097**|
> > > |16|.247|.060|**.050**|.077|.092|.056|.053|
> > >
> > > **LegalBench (abercrombie)**
> > >
> > > |m|Pop.Mean|Sample|IRT|Anchor Points|DKPS|DKPS+IRT|Ens(DKPS+IRT)|
> > > |---|---|---|---|---|---|---|---|
> > > |1|.168|.421|.385|.364|.170|.170|**.170**|
> > > |4|.168|.185|.165|.184|.103|.102|**.100**|
> > > |16|.168|.090|.082|.073|.057|**.055**|.056|
> > >
> > > These results confirm that DKPS is generally complementary to existing methods; across both tasks at m=1 and m=4, DKPS consistently leads. The one case where a baseline leads outright is MATH at m=16, where IRT (.050) outperforms DKPS+IRT (.056). We expect this gap to narrow or flip with more reference models, and that the complementarity will hold or improve in that setting.
> > >
> > > We note that Anchor Points' underperformance relative to IRT and DKPS is expected. In particular, the method was designed in a setting with per-class prediction probabilities / rankings and — to our knowledge — has not been explored or optimized for the purely black-box setting. Our implementation uses a naive black-box signal. Given this, we do not plan to include it in the paper.
> > >
> > > We will add IRT, DKPS+IRT, and Ens(DKPS+IRT) for all datasets to the main result table (Table 1) to support our hypothesis that DKPS is complementary to "competitors". We will add descriptions of the implementation details in a dedicated appendix section.
> > >
> > > ---
> > > We appreciate your continued engagement! Please let us know if you have any remaining concerns.

---

### Official Review · Reviewer_LG8s · 2026-03-13

**Soundness:** 2
**Presentation:** 3
**Significance:** 3
**Originality:** 2
**Overall Recommendation:** 4
**Confidence:** 3

**Summary:**

The paper studies query-efficient benchmark evaluation for generative models using cached responses from previously evaluated models. The authors construct model representations in DKPS from response embeddings on a small subset of benchmark queries, and use regression in this space to predict the full benchmark score of a target model. The paper provides both theoretical analysis and empirical evaluations showing that the proposed method can achieve more accurate benchmark score estimation than directly using subset scores under limited query budgets.

**Compliance With Llm Reviewing Policy:**

Affirmed.

**Key Questions For Authors:**

Please see weaknesses

**Limitations:**

yes

**Strengths And Weaknesses:**

Strengths:
S1: The problem is important
S2: The proposed method can improve benchmark evaluation efficiency.
S3: The experiments show clear gains at low query budgets.

Weaknesses:
W1. In Section 3, the theoretical result establishes query-efficiency for nearest-neighbor regression in DKPS. However, the empirical method in Section 4 instead uses ordinary least squares regression combined with an ensemble with subset scores. Since the theoretical guarantee does not apply to this estimator, the theory does not directly justify the practical method. The authors could clarify why this estimator is chosen in practice and whether stronger theoretical justification can be provided for the empirical method.

W2. In Section 4, constructing the DKPS representation requires computing distances between models based on responses to the same subset of queries, meaning all reference models must share a common query set. The experiments therefore restrict to the maximal subset of models with overlapping queries, while partial-overlap scenarios are left for future work. This assumption may limit applicability in realistic benchmark settings where models are evaluated asynchronously on different query subsets. Please discuss how restrictive this assumption is in practice and whether the method could be extended to partial-overlap settings.

W3. The empirical method in Section 4 relies on several design choices that are not systematically analyzed, including the embedding function, the DKPS dimension d, and the ensemble weight \alpha = m/M used to combine DKPS predictions with subset scores. The paper shows that embedding choice noticeably affects results at small query budgets, suggesting these factors may influence performance. Additional analysis on the sensitivity of the method to these design choices would strengthen the evaluation.

W4. The theoretical analysis in Section 3 assumes deterministic response-level scoring, where each model-query pair has a fixed score. However, many modern evaluation pipelines rely on stochastic LLM-as-judge scoring, where evaluation outputs may vary across runs. The current theoretical framework does not cover this setting.

W5. In Section 4, the experiments mainly compare against Sample Score and Population Mean baselines. The paper does not include comparisons with other approaches designed to reduce benchmark evaluation cost, such as subset-selection or adaptive evaluation methods discussed in related work. It would be helpful to include at least representatives of these baselines, or provide a clear justification for not including them.

W6. The approach relies on cached responses from previously evaluated models to construct the DKPS space. If the reference model set is small or lacks diversity, the DKPS geometry may not accurately capture relationships between models, which could reduce prediction accuracy. Further analysis of how the method performs under limited or biased reference model coverage would be helpful.

W7. In Section 4, the paper proposes an offline query selection strategy based on maximizing the R^2 fit on reference models. However, this strategy is mainly evaluated on the MATH benchmark, and it is unclear whether the same heuristic generalizes well across other benchmarks or task types (e.g., QA or translation tasks)

---

> ### Author Rebuttal · Authors · 2026-03-27
>
> Thank you for taking the time to provide a detailed review of our paper. We think that your summary is a fair representation of the work. We address the weaknesses by your labeling (W1-W7).
>
> > W1
>
> Please see our response to Reviewer G1YB.
>
> > W2
>
> In practice the size of the common query set is substantial. For within-organization benchmarking the query set for each model is often forced to be the common query set. This is particularly true when benchmarking in a domain where high quality queries are expensive. The largest common query set in the public benchmarking setting is likely the smallest one will observe, especially if the public benchmark accepts self-reporting.
>
> As discussed in the Limitations, the method can be extended to handle non-overlapping query sets. Given the additional complexities that this would introduce to the current analysis (the ratio & relationship between overlapping queries to non-overlapping queries, the “best” way to combine the types of information, etc.), we decided to leave investigations along this line to future work.
>
> > W3
>
> We have since run the experiments with different DKPS $d$ and different $\alpha$. The results are detailed in the tables below. We fix n=ALL for both results. We fix $d=8$ for the interpolation result and $\alpha = m/M$ for the $d$ result. Due to character constraints we only included the results for LegalBench and MATH. The results for MedQA and WMT-14 are qualitatively similar.
>
> ## LegalBench
>
> |m|d=1|d=2|d=8|d=16|d=32|
> |--:|----:|----:|----:|-----:|-----:|
> | 1 | 0.149 | 0.140 | 0.139 | 0.148 | 0.178 |
> | 4 | 0.134 | 0.121 | 0.101 | 0.108 | 0.122 |
> | 16 | 0.084 | 0.071 | 0.058 | 0.059 | 0.068 |
>
>
> ## MATH
>
> |m|d=1|d=2|d=8|d=16|d=32|
> |--:|----:|----:|----:|-----:|-----:|
> | 1 | 0.189 | 0.167 | 0.146 | 0.154 | 0.183 |
> | 4 | 0.138 | 0.123 | 0.108 | 0.113 | 0.127 |
> | 16 | 0.108 | 0.103 | 0.088 | 0.092 | 0.095 |
>
> Moderately sized $d$ appear sufficient for high-quality prediction. Performance suffers for all $m$ when $d$ is too small or large (relative to the number of ref. models). Now the effect of $\alpha$:
>
> ## LegalBench
> | m | m/M | 0 | 0.1 | 0.5 | 0.8 |1|
> |--:|----:|--:|----:|----:|----:|----:|
> | 2 | 0.122 | 0.123 | 0.122 | 0.162 | 0.223 | 0.267 |
> | 4 | 0.100 | 0.101 | 0.100 | 0.121 | 0.156 | 0.181 |
> | 16 | 0.056 | 0.058 | 0.057 | 0.061 | 0.072 | 0.082 |
>
> ## MATH
>
> |m|m/M|0|0.1|0.5|0.8|1|
> |--:|----:|--:|----:|----:|----:|----:|
> | 2 | 0.122 | 0.125 | 0.120 | 0.137 | 0.183 | 0.225 |
> | 4 | 0.101 | 0.108 | 0.101 | 0.102 | 0.128 | 0.155 |
> | 16 | 0.061 | 0.088 | 0.080 | 0.057 | 0.054 | 0.061 |
>
> While $m/M$ is not the best for any given $m$, it is near-best for all $m$ and is a solid default.
>
> We plan to include extended versions of these studies in the appendix. We would also appreciate suggestions on how to better analyze the effect of the embedding function.
>
> > W4
>
> As mentioned in the Limitations section, the theoretical framework can be extended to stochastic scoring functions. The particular extension will depend on the set of assumptions one is willing to make. For example, if each score from the LLM-as-a-judge is of the form $y+e_{response}+e_{judge}$ with each $e_{response}$ and $e_{judge}$ iid from model/judge-specific distributions, then repeatedly querying the target model and repeatedly scoring from the judge will recover the current setting as the number of responses and scores increase. We leave investigations into how properties of the judging distribution affect the query-efficiency to future work and will note this particular extension and its relationship to our current framework in the appropriate paragraph in the Limitations section.
>
> > W5 / W7
>
> Please see our response to Reviewer FvXQ.
>
> > W6
>
> While we do not study the effect of biased model coverage explicitly, we do study the effect of $n$ (20, 50, ALL) on the performance of the DKPS-only method (Figure 2). In general, the more models the better and DKPS-only can outperform the Sample Score for query budgets that are O(10) even for a modest ($n=20$) number of models. Further, our existing empirical results follow a “Leave-One-Family-Out” protocol – effectively ensuring that our results are not reliant on models that are artificially similar to the target model.
>
> With that said, we report the mean min/max/median/IQR performance of the Ensemble method  (with $n=20, d=8, \alpha=m/M$) conditioned on a query set of size $m=5$. (The “mean” is across models; the min/max/etc is with respect to 100 randomly selected collections of reference models.) :
>
> | dataset | min | Q25 | med | Q75 | max |
> |:--------|----:|----:|----:|----:|----:|
> | LegalBench | 0.010 | 0.073 | 0.118 | 0.181 | 0.427 |
> | MATH | 0.008 | 0.076 | 0.122 | 0.203 | 0.420 |
>
> With $n=20$, the performance depends highly on the reference models. Please let us know if the above result sufficiently addresses W6 – we plan to include an extended version of it as a figure in the main text with the additional space.

---

> > ### Author Rebuttal · Reviewer_LG8s · 2026-04-01
> >
> > Thanks for the rebuttal.  I am raising the score to a positive score.

---

> > > ### Author Response · Authors · 2026-04-03
> > >
> > > Thank you for acknowledging our rebuttal.
> > >
> > > As mentioned in our discussion with Reviewer G1YB, we have been able to address your Weakness 5 in the time between the initial rebuttal submission and now.
> > >
> > > As always, we appreciate your continued engagement!

---

### Official Review · Reviewer_FvXQ · 2026-03-23

**Soundness:** 4
**Presentation:** 3
**Significance:** 3
**Originality:** 3
**Overall Recommendation:** 5
**Confidence:** 3

**Summary:**

This paper uses the "data-kernel perspective space" to embed model queries and responses in order to predict the performance of an unseen model on a query set based on a subset of cached responses from other models. The DKPS approach allows prediction of model accuracy with only black-box access to model responses, as compared to prior work that requires knowledge of model internals or metadata.

**Compliance With Llm Reviewing Policy:**

Affirmed.

**Key Questions For Authors:**

If there are any results comparing this method to previous (even if non-black-box) methods, those would be informative.

**Limitations:**

The limitations are discussed quite thoroughly.

**Strengths And Weaknesses:**

Strengths:
- The work is well-motivated and proposes a clearly described solution to the problem of predicting model outcomes with limited evaluation query budget.
- The proposed method is black-box, which opens up the use of publicly available model results/evaluated queries.
- The ensemble method effectively allows a tradeoff between the DKPS method and the sample score method in the regimes where each of these is effective. This is further supported by the theoretical results, which are clearly presented and interpreted.


Weaknesses:
- Even though the current work targets black-box access to models, it would benefit the work to include at least one comparison to prior work on benchmarking to contextualize the work beyond just the baselines chosen by the authors. This would give context for how much accuracy/predictive power is lost (if any) when using only black-box access to model responses.


Nits: It seems relevant to discuss the line of work on datamodels, which targets a quite similar problem to what the authors target here https://arxiv.org/pdf/2202.00622 (albeit is far more training-intensive).

---

> ### Author Rebuttal · Authors · 2026-03-27
>
> Thank you for taking the time to provide a thorough review. We think your summary is a fair representation of our work. We appreciate your comments on our description of the problem and our results and your acknowledgement that black-box methods are important so that *all* models can be considered under the same framework. We address your mentioned weaknesses (and nit) below.
>
> > If there are any results comparing … would be informative.
>
> For this rebuttal period we opted to spend resources improving the exploration of the proposed method by investigating its sensitivity to different embedding dimensions, to different interpolations of DKPS and Sample Score, and worst-case analysis for a small number of models. With that said, we will better contextualize our work with respect to the methods references and improve the description of how to use the proposed method when additional / alternative efficient benchmarking methods are available. We will also provide a stronger intuition as to why we expect the efficiency gains to be complementary.
>
> > It seems relevant to discuss … https://arxiv.org/pdf/2202.00622 (albeit is far more training-intensive).
>
> Thank you for pointing us to the datamodels paper. It is relevant in that they study how structural assumptions on a collection of hypotheses can be used to predict model outputs. In an abstract sense, you can think of a model output as a very particular model-level covariate – we will add a sentence describing this relationship in the related works.
>
> Please let us know if you have any additional questions or concerns!

---

> > ### Author Rebuttal · Reviewer_FvXQ · 2026-04-03
> >
> > I will maintain my score because I am convinced of the empirical value of this method. That said, I did not carefully read the theoretical parts in my initial review and would defer to the reviewers who engaged more closely with the theory on whether those issues materially impact the empirical usefulness of the work.

---

### Decision · Program_Chairs · 2026-04-30

**Decision:**

Accept (regular)

**Comment:**

This paper proposes a query-efficient benchmark evaluation framework using the Data Kernel Perspective Space (DKPS), which embeds models based on their cached responses to predict a target model's full benchmark score from a small query subset. The problem is practically important and growing in relevance, and the Leave-One-Family-Out evaluation protocol is a strong methodological choice.

Reviewer opinions are mixed with an active line of rebuttal.

The most critical empirical gap — the absence of comparisons to existing efficient benchmarking methods — was addressed in the follow-up rebuttal with IRT and Anchor Points results, confirming that DKPS is complementary: it leads at low query budgets and DKPS+IRT is the strongest overall combination. These results substantially strengthen the paper's contribution.

Remaining concerns are also discussed (some are resolved), including a theory-practice gap (the theoretical guarantees cover k-NN while the practical method uses OLS), limited novelty relative to the existing DKPS line of work, and some notation and presentation issues in the theoretical sections.

The authors are strongly encouraged to incorporate all follow-up experiments into the final version, clarify the relationship to prior DKPS work, and improve the theoretical presentation.